# Mucus production, host-microbiome interactions, hormone sensitivity, and innate immune responses modeled in human cervix chips

Zohreh Izadifar[1,5], Justin Cotton [1], Siyu Chen [2], Viktor Horvath [1], Anna Stejskalova [1], Aakanksha Gulati[1], Nina T. LoGrande[1], Bogdan Budnik[1], Sanjid Shahriar [1], Erin R. Doherty[1], Yixuan Xie [2], Tania To[1], Sarah E. Gilpin[1], Adama M. Sesay[1], Girija Goyal[1], Carlito B. Lebrilla [2] & Donald E. Ingber [1,3,4] ✉

Modulation of the cervix by steroid hormones and commensal microbiome play a central role in the health of the female reproductive tract. Here we describe organ-on-a-chip (Organ Chip) models that recreate the human cervical epithelial-stromal interface with a functional epithelial barrier and production of mucus with biochemical and hormone-responsive properties similar to living cervix. When Cervix Chips are populated with optimal healthy versus dysbiotic microbial communities (dominated by *Lactobacillus crispatus* and *Gardnerella vaginalis*, respectively), significant differences in tissue innate immune responses, barrier function, cell viability, proteome, and mucus composition are observed that are similar to those seen in vivo. Thus, human Cervix Organ Chips represent physiologically relevant in vitro models to study cervix physiology and host-microbiome interactions, and hence may be used as a preclinical testbed for development of therapeutic interventions to enhance women's health.

In most women, the optimal healthy state of the lower reproductive tract is characterized by a symbiotic relationship with a cervico-vaginal microbiome composed of a stable community of bacteria dominated by *Lactobacillus crispatus*[1]. When a dysbiotic state develops, such as bacterial vaginosis (BV), this composition shifts and the *L. crispatus* community is replaced by a non-stable community of diverse 'non-optimal' anaerobes with an increased presence of more pathogenic bacteria, such as *Gardnerella vaginalis* and fewer *L. crispatus*[2]. BV is a common clinical condition that affects >25% of reproductive-age women globally[3] and contributes to significant adverse health outcomes including increased risk of acquiring human immunodeficiency

virus (HIV) and other sexually transmitted infections[4,5] as well as higher rates of spontaneous abortion, pre-term birth, miscarriage, low birth weight, pelvic inflammatory disease, and postpartum endometritis[6–8]. Current therapeutic options include antimicrobial and probiotic regimens and biofilm disruption[2], however, they are of limited value and have a high failure rate that results in frequent recurrence (up to 50% within 1 year)[9]. Although the BV syndrome was first described over a century ago, the mechanisms that underlie this condition and the casual linkages between dysbiosis and adverse health outcomes still remain unknown, which has prevented development of more effective treatment and prevention strategies[3,10]. This is in part because most of

[1]Wyss Institute for Biologically Inspired Engineering, Harvard University, Boston, MA 02215, USA. [2]Department of Chemistry, University of California Davis, Davis, California, Davis, CA 95616, USA. [3]Vascular Biology Program, Boston Children's Hospital and Department of Pathology, Harvard Medical School, Boston, MA 02115, USA. [4]Harvard John A. Paulson School of Engineering and Applied Sciences, Cambridge, MA 02134, USA. [5]Present address: Urology Department, Boston Children's Hospital, Harvard Medical School, Boston, MA 02115, USA. ✉e-mail: don.ingber@wyss.harvard.edu

our current knowledge and therapeutic development approaches are based on results of studies of microbial cultures[11,12], metagenomic analyses[1,13,14], in vitro[10], or animal[15–17] models that fail to faithfully mimic the complex tissue microenvironment and mucus layer of the human reproductive tract.

The mucus layer that lines the epithelial surfaces of the entire lower reproductive tract is increasingly recognized as a key contributor to fertility and pregnancy, as well as reproductive and genital health outcomes[18–20], although the underlying mechanisms remain poorly understood. Human cervix is constituted of two main regions, which include the endocervix that is lined by a columnar epithelium and extends from the lower opening of the uterus to the external opening of the cervical canal and the ectocervix which is covered by a glycogen-rich, stratified squamous epithelium that stretches from the canal opening to the vagina[21]. The structure, composition, molecular signaling[15,22], and physiological features of cervical epithelium, including mucus properties, change dramatically with hormonal (e.g., sex steroids) and environmental (i.e. pH, oxygen tension) alterations, and this plays a critical role in the reproductive health and function. Changes in bacterial composition within the mucus layer also can reciprocally stimulate host transcriptional and immune responses[23–26], thereby further altering mucus composition as well as the associated vagino-cervical microbiome[14]. For instance, changes in sex hormone levels during menarche, menses, pregnancy and menopause alter both the vaginal microbiome[27,28] and mucus properties[29,30]; however, it is currently impossible to determine whether these changes in mucus alter the microbiome or vice versa in vivo in the human cervix. Little is also known about differences between endo- and ecto-cervical physiology. Thus, to tease out the dynamic interactions between different contributing factors and thereby, potentially provide new approaches to reduce the burden of BV and its associated health and economic sequelae among millions of women around the world, it would be helpful to have an in vitro model of the human cervical mucosa that can recapitulate the physiology and complexity of these different cervical microenvironments yet allow for controlled perturbation and analysis of how these various potential contributors alter mucus biology and thereby influence reproductive health and disease.

Although current in vitro culture systems, such as two dimensional (2D) planar cultures[31,32], Transwell inserts with porous membranes[14,33], and cultured three dimensional (3D) engineered constructs developed using rotating wall vessels[23,34,35], have been used to study human cervical epithelial cell interactions with bacteria and microbicides as well as innate immune responses[31,32], they fail to recapitulate the physiological tissue-tissue interfaces, barrier properties, mucus production, molecular signaling, secretory profile, and dynamic host-microbiome interactions that are observed in human cervix in vivo. These culture models also largely rely on cervical epithelial cell lines or cervical cancer cells[23,36,37], with few advanced versions that use a commercial source of primary human epithelial cells cultured in Transwell inserts[38,39]. The cervical mucus that plays such a key role in maintaining healthy host-microbiome interactions has not been modeled effectively in vitro. It is equally difficult to study interactions between human cervical epithelium with living microbiome for extended times (>1 day) in these static cultures because the microbes overgrow and kill the human cells. Because of these limitations, existing models of the human cervix cannot be used to analyze complex host-microbiome interactions that are observed in vivo for meaningful preclinical screening of new therapeutics and other clinical interventions in diseases of the reproductive tract such as BV.

Importantly, organ-on-a-chip (Organ Chip) microfluidic culture technology that enables investigators to recreate organ-level structures including tissue-tissue interfaces that experience dynamic fluid flow in vitro offers a potential way to overcome these challenges[40,41]. For example, we recently described a human Vagina Chip lined by primary human vaginal epithelium and underlying stromal fibroblasts that faithfully recapitulates the structure and function of the human vaginal mucosa[42]. While, the cervix is a major source of the mucus that normally serves as a protective covering over the surface of the cervical and vaginal epithelium that interacts directly with resident bacterial flora in the entire lower female reproductive tract[20,43,44], existing in vitro models do not faithfully mimic cervical mucus production or physiology.

Here, we describe the development of an Organ Chip model of the human cervical mucosa (Cervix Chip) lined by primary human cervical epithelium interfaced with human cervical stromal fibroblasts. To benchmark our results against prior in vitro models of human cervix[38,39], we use a commercially available source for the primary human cervical epithelial cells used in previous studies that contains a mixture of endo- and ecto-cervical epithelial cells. When we culture these cells under different dynamic flow conditions on-chip, we are able to create Cervix Chips that preferentially express an ecto- or endo-cervical phenotype and which respond to hormonal, environmental, and microbial cues in a manner similar to that observed in vivo. These Cervix Chips also produce mucus that more closely resembles cervical mucus in vivo than that produced in static Transwell cultures with the same cells and they enable study of host-microbiome interactions for multiple days in vitro.

## Results
### Engineering of mucus-producing human Organ Chip models of endo- and ecto-cervix

Organ Chip models of the human cervical mucosa were developed using a commercially available microfluidic chip (Emulate Inc., Boston MA) composed of an optically clear silicone rubber polymer that contains two microchannels separated by a thin, porous membrane of the same material (Fig. 1a). To recreate the human cervical epithelial-stromal interface on-chip, the commercially available human primary cervical epithelial cells, similar to those used in prior in vitro cervix models[38,39], were cultured on the top surface of the porous membrane which was pre-coated with extracellular matrix (ECM) and primary human cervical stromal fibroblasts, isolated from healthy cervical tissue, were cultured on the lower surface of the same membrane (Fig. 1a, b). The commercially acquired cervical epithelial cells did not contain any technical information about the specific anatomical region from which the cells were isolated, but after characterizing them we discovered that they contained a mixed population of endo- and ecto-cervical epithelial cells (Supplementary Fig. 1a). Given the limited availability of healthy primary cervical epithelial cells and our desire to benchmark our results against past studies with static culture models, we continued our chip development effort with this mixed cell population.

In past human Organ Chips studies, incorporating relevant mechanical cues specific for the organ microenvironment (e.g., cyclic deformations in lung and intestine, low levels of fluid flow in kidney, etc.) were found to be crucial for effectively replicating organ-level physiology and pathophysiology[45–47]. In the human cervix, mucus is continuously produced and cleared from the epithelium surface as it flows slowly to the vagina by mucociliary transport and through the effects of gravity. Because the physiological mucus flow conditions in vivo are not known, two different flow regimens were tested on the chip: continuous perfusion of the upper channel with epithelial growth medium (Continuous) at 40 µl/hr or intermittent perfusion in which the same medium was flowed for 4 hrs at 30 µl/hr followed by 20 h of static culture (Periodic) to mimic gentle frequent and infrequent mucosal discharge flow, respectively, which can occur in vivo. In both conditions, the lower channels were perfused continuously with stromal growth medium. After 5 days of cell expansion, the apical medium was replaced with a customized Hank's Buffer Saline Solution (HBSS) at pH ~ 5.4 with lower buffer salts and no glucose (LB/-G) to simulate the slightly acidic pH of the cervix[48], while the stromal growth medium was

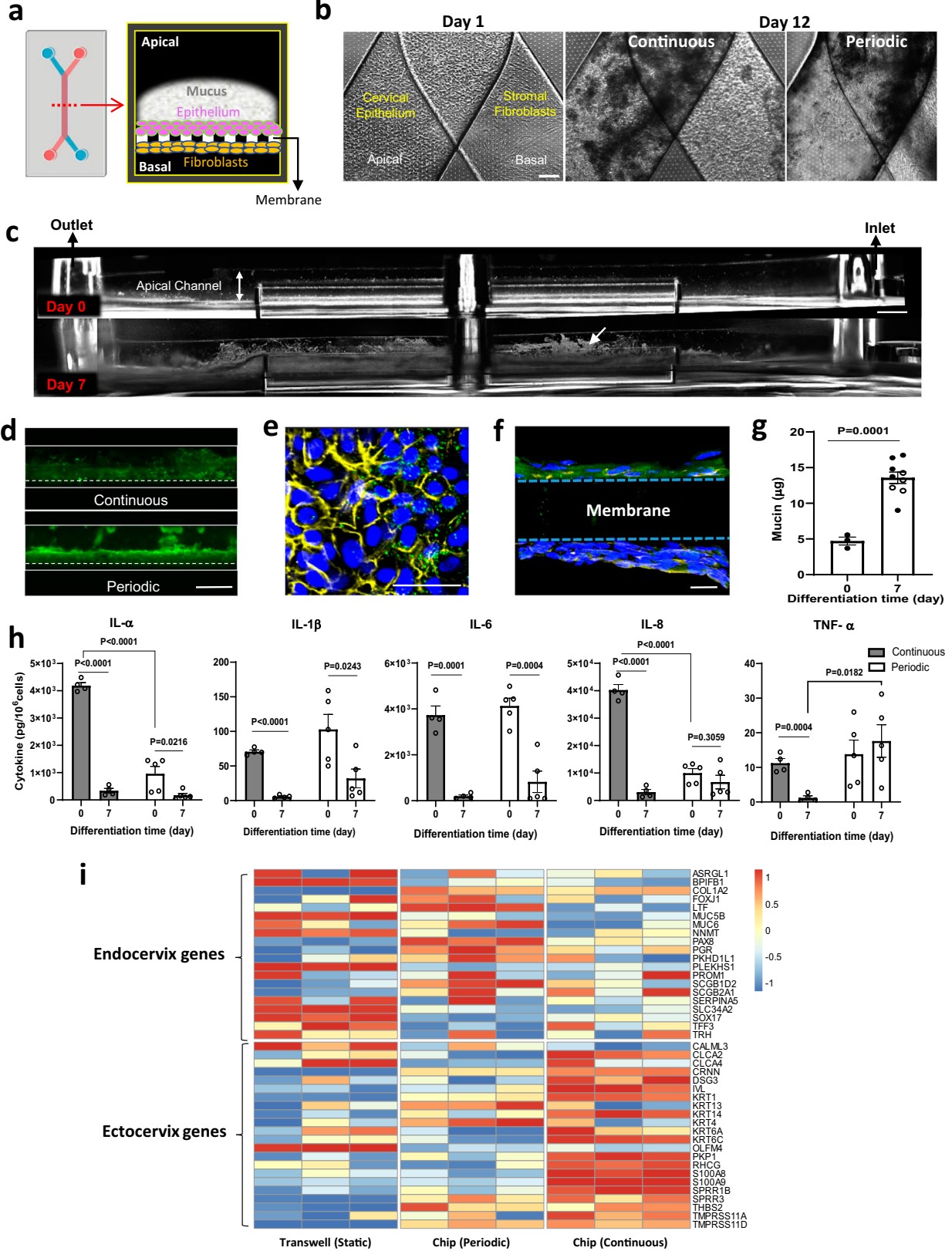

replaced with differentiation medium containing the female sex hormone estradiol-17β (E2) at a dose (5 nM) that represents the peak estrogen level experienced during the follicular phase of the menstrual cycle[49]. Culture under these conditions were designed to promote cell differentiation (i.e., expression of tissue-specific behaviors) that was continued for 7 additional days (until day 12), and then they were

inoculated with microbial consortia in studies of host-microbiome interactions.

Both the epithelial and stromal cells formed confluent monolayers within 1 day after seeding on-chip, and dark amorphous regions of what appeared to be mucus accumulation were detected above the epithelium under both flow conditions when viewed from above by

**Fig. 1 | Development and characterization of human ecto-and endo-cervix chips. a** Schematic top (left) and cross-sectional views (right) of the dual channel microfluidic organ chip lined by human cervical epithelium interfaced across an ECM-coated porous membrane with human cervical fibroblasts. **b** Phase-contrast microscopic view from above of a Cervix Chip lined by cervical epithelium and fibroblasts on the apical and basal sides of the chip porous membrane, respectively, on day 1 (left) as well as after 12 days of culture under continuous (middle) or periodic (right) flow (bar, 200 μm). **c** Side view, dark field images of living Cervix Chip hours after cell seeding day 0 (top) and after 7 days of differentiation (bottom). White arrow indicates light reflective mucus accumulating above the differentiated epithelium (bar, 1 mm). **d** Fluorescence microscopic side views of mucus layers in live Cervix Chip cultures stained with fluorescent wheat germ agglutinin (WGA) (green) on day 7 of differentiation under continuous (top) and periodic (bottom) flow regimens (bar, 1 mm). **e** Immunofluorescence microscopic view from above of the cervical epithelium stained for mucin 5B (MUC5B, green), F-actin (yellow), and nuclei with Hoechst (blue) (bar, 50 μm) stained at day 7 of differentiation. **f** Immunofluorescence microscopic vertical cross sectional view of the chip showing the cervical epithelium stained for MUC5B (green) overlaying the porous membrane, the underlying layer of fibroblasts stained for vimentin (yellow), and Hoechst-stained nuclei (blue) (bar, 20 μm). **g** Quantification of the total mucin content produced by Cervix Chips cultured under continuous (black symbols) or periodic (white symbols) flow measured at days 0 and 7 of differentiation. **h** Cytokine proteins in effluents of Cervix Chips cultured under continuous (gray bars) or periodic flow (white bars) on day 0 versus 7 of differentiation. **i** RNAseq analysis of genes expressed in cervical epithelial cells from three different donors differentiated in static Transwells or Cervix Chips under periodic or continuous flow. Expression levels of the signature genes associated with the endo- and ecto-cervical epithelial cells[59] were compared using Z-scores calculated per donor across samples for the three flow conditions, emphasizing common trends among samples of different donors (colors represent gene expression levels; each column represents one of the three donors). Data represent the mean ± s.e.m.; $n = 3$ (periodic flow) and 5 (continuous flow) (**g**), 4 (continuous flow) and 5 (periodic flow) (**h**), and 3 (**i**) experimental chip replicates. Micrographs in (**b**–**f**) are representatives of three separate experiments. Source data and statistical tests are provided as a Source Data file. Left image in (**a**) was created with BioRender.com.

phase contrast microscopy (Fig. 1b). Similarly, when viewed from the side by dark field microscopic imaging of live cultured chips (which allows direct visualization of mucus on-chip)[40] after 7 days of differentiation (culture day 12), a fuzzy, white, light scattering material had accumulated above the epithelium nearly filling the apical channel (Fig. 1c). This was observed using both continuous and periodic flow regimens and the material stained positively with fluorescently labeled Wheat Germ Agglutinin (WGA) lectin, which preferentially binds to glycans of the major MUC5B mucus protein[50] (Fig. 1c, d). This was further validated using immunofluorescence microscopic imaging of chips stained for MUC5B and F-actin which confirmed the presence of this primary gel-forming mucin type of human cervical mucus[43,51] within the cervical epithelial cells when viewed from above (Fig. 1e) or from the side in vertical cross sections (Fig. 1f). This study also confirmed the presence of vimentin-containing fibroblasts and collagen fibers (visualized using Second Harmonic Generation microscopy) on the underside of the membrane (Fig. 1f, Supplementary Fig. 1b) as well as expression of cytokeratin 18 (CK18), estrogen receptor (ER), and progesterone receptor (PGR) in the cervical epithelial cells stained on day 7 of differentiation (Supplementary Fig. 1b), which mimics their expression observed in the cervix in vivo[52,53].

Histological staining of frozen vertical cross sections of the Cervix Chips showed that the epithelial cells cultured under continuous flow formed a stratified epithelium with multiple (>6) layers of flattened cells that is reminiscent of ectocervix, while under periodic flow, the epithelial cells formed fewer layers (2–4) and exhibited a range of forms from cuboidal to more flattened (Supplementary Fig. 1b). While epithelial cells have the potential to adopt characteristics of mesenchymal cells through an epithelial-mesenchymal transition, constant exposure of Cervix Chips to sex hormones and dynamic fluid flow prevented this transdifferentiation from occurring in these cervical cells. This was confirmed by continued detection of cervical epithelium-specific features, including high levels of MUC5B production and expression of cytokeratin 18 as well as estrogen and progesterone receptors that are not typically associated with mesenchymal cells. The presence of cervical stroma also enhanced mucus production compared to culture conditions when stromal cells were absent (Supplementary Fig. 1c).

Quantification of mucus accumulation using a colorimetric Alcian Blue assay[54] confirmed that there was a significant (~3-fold) increase in the total amount of mucin after 7 days of differentiation on chip under both continuous and period flow conditions (Fig. 1g). Quantitative analysis of immunofluorescence images of vertical cross-sections of continuous culture chips stained for MUC5B independently showed a similar 3-fold increase in the epithelial surface area covered by MUC5B-containing mucus secretions and this was accompanied by a

concomitant decline in cell proliferation as measured by decreased expression of the proliferation marker Ki67 (from ~60% to < 10% labeling index) over more than a week-long differentiation time course (Supplementary Fig. 1d). Real time quantitative PCR (RT-qPCR) of the differentiated epithelial cells in the periodic flow chip also demonstrated significant upregulation of genes encoding *MUC5B* as well as *MUC4* and secretory leukocyte peptidase inhibitor (*SLPI*) at day 7 of differentiation compared to when the same cells were cultured in conventional 2D cultures using the same medium where these genes were barely expressed (Supplementary Fig. 2a).

Importantly, reconstitution of this epithelial-stromal interface similar to that observed in vivo was accompanied by establishment of a stable tissue barrier, as demonstrated by measuring real-time transepithelial electrical resistance (TEER) using electrodes that were integrated into custom made Organ Chips of the same design and composition, which enable continuous and quantitative measurement of barrier function over many days in culture[55]. The cervical mucosa formed on-chip maintained a barrier with high electrical resistance ranging from 150–800 ($\Omega$.cm²), which was significantly greater than that generated when the same cervical and stromal cells were seeded on a similar permeable membrane in a commercial static Transwell insert (50–350 $\Omega$.cm²) using the same medium conditions (Supplementary Fig. 2b). In both the chip and Transwell models, the cervical tissue barrier maintained higher levels of TEER during the expansion phase and the first 2 days of differentiation, which then decreased and stabilized at a lower level. Interestingly, the presence of cervical stroma in the culture system was also found to be essential for formation of an effective epithelial barrier as TEER was lower in its absence (Supplementary Fig. 1c).

We then analyzed the innate immune status of these Cervix Chips cultured under both periodic and continuous flow by quantifying cytokine protein levels in the effluents of the upper epithelial channels 24 h after 7 days of differentiation. These studies revealed that the induction of tissue-specific epithelial cell specialization and mucus accumulation was accompanied by a significant reduction in the levels of the pro-inflammatory cytokines. IL-1α, IL-1β, and IL-6 levels decreased regardless of whether cells were cultured under continuous or periodic flow; however, only exposure to continuous flow reduced IL-8 and TNF-α levels as well (Fig. 1h). The higher levels of IL-8 and TNF-α observed in the periodic flow chip are consistent with previous reports of elevated IL-8 secretion induced through TNF-α stimulation by cyclic mechanical stimulation in reproductive tract epithelial cells[56–58].

Interestingly, when we carried out RNA-seq analysis of the cervical epithelium under the different flow conditions on-chip, we found that cells cultured under continuous flow exhibited significant

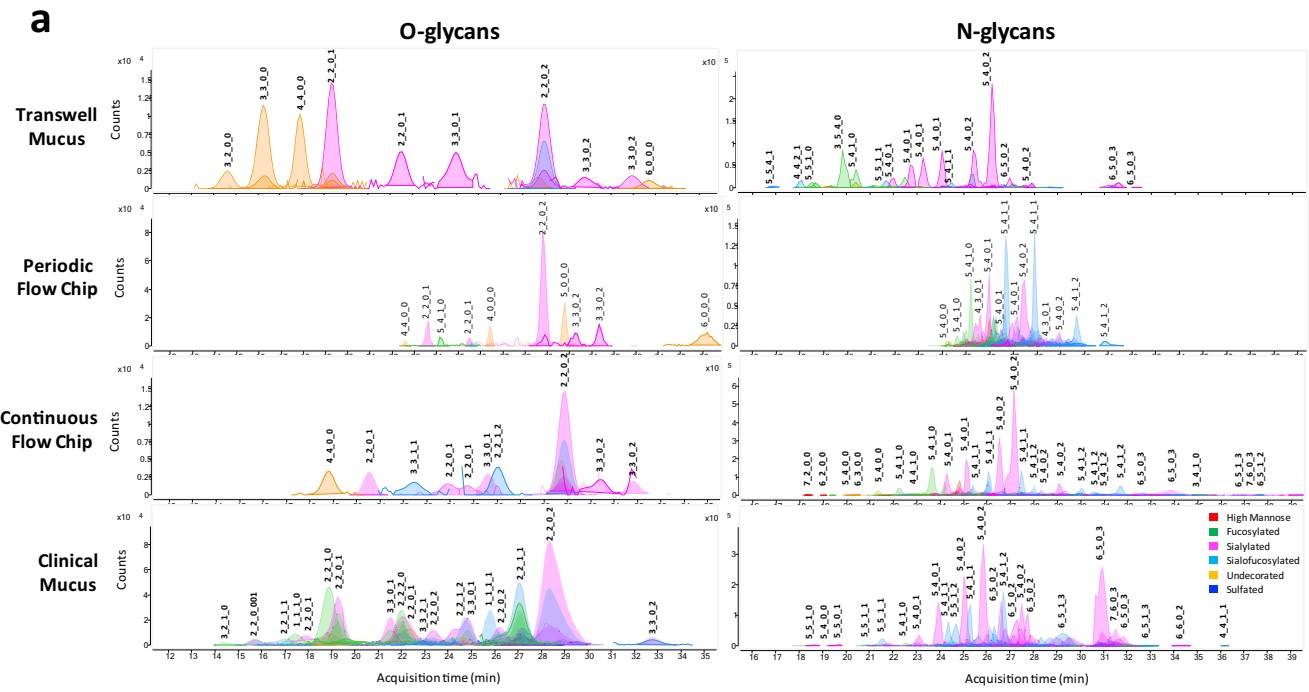

**Fig. 2 | Recapitulation of the compositional properties of cervical mucus.**
**a** Glycomic analysis showing representative O- and N-glycan profiles of mucus produced by cervical epithelial cells cultured in Transwells or Cervix Chips exposed to periodic or continuous flow compared to a clinical sample of human mucus. The compound peaks are color coded to show the glycan subtypes and numbered to represent the glycan molecules structures: (#_#_#_#) format represents number of

core structure type 1_number of core structure type 2_number of glycan subtype 1_number of glycan subtype 2. **b** Tables showing the most abundant O- and N-glycan subtypes observed in the mucus profiles of human clinical samples compared with Cervix Chips exposed to periodic or continuous flow (combined peaks) versus Transwell cultures.

upregulation of genes associated with ectocervical epithelial cell phenotype[59] including multiple different keratinization-associated genes (cytokeratin; *KRT1, 14, 13, KRT6A, KRT6C,* involucrin; *IVL*), whereas the same cells cultured in the same medium under periodic flow or under static conditions in Transwell inserts upregulated genes more closely associated with the endocervical epithelial phenotype[59], such as secretory-associated (mucin; *MUC5B* and *MUC6, PAX8*) and ciliogenesis-related (cilia protein; *FOXJ1*) genes (Fig. 1i). The genes shown in Fig. 1i were selected because they have been previously reported to be differentially expressed in the transcriptome of human ectocervix versus endocervix in vivo[59]. As the primary cells we used contain a mixture of both endo- and ecto-cervical cells, these different flow conditions likely either promote growth of only one subclass of the cervical epithelial cells or they have differential effects on cervical epithelial cell specialization. This finding that exposure to continuous flow results in preferential expression of the ectocervical phenotype

compared to periodic flow which exhibited more endocervical features was confirmed in chips created with cells isolated from three different human donors using a gene set scoring analysis method[60] (Supplementary Fig. 2c). Interestingly, while there was some donor-to-donor variability, cells behaved differently when cultured in static Transwells where they expressed significantly more endocervical features even compared with periodic flow that also experienced intermittent static periods, while the epithelial cells in continuous flow chips more closely resembled the ectocervical phenotype observed in vivo (Supplementary Fig. 2c).

These results suggests that the continuous flow condition preferentially promotes ecto-cervical epithelial growth or specialization, while periodic flow that includes static periods and conventional Transwell cultures push these epithelial cells towards an endo-cervical phenotype. However, it is important to note that the Cervix Chip phenotype under periodic flow is distinct from that displayed by the

same cells in the same medium in static Transwells. For example, while *MUC5B* appears to be expressed at a similar level under both conditions, *MUC4* gene expression in the cervical epithelium was significantly higher on-chip (Supplementary Fig. 2a). Cervix Chips cultured under periodic flow also exhibited enhanced epithelial barrier integrity compared to Transwells (Supplementary Fig. 2b). There were also differences in the cell morphology in Transwells versus chips with the epithelium forming a more regular polygonal cell layer and the stroma exhibiting greater alignment in the Cervix Chip cultured under periodic flow (Supplementary Fig. 2d). Finally, transcriptomic analysis revealed that collagen genes expressed in the Cervix Chip are more similar to those detected in endo- and ecto-cervical tissues in vivo than those expressed when the same cells are cultured in static Transwell cultures (Supplementary Fig. 2e).

## Analysis of cervical mucus chemistry and physiology on-chip

Mucin proteins are heavily glycosylated (40–80%) and their composition and glycomic structure play an important role in fundamental inter- and intra-cellular processes and immune modulation functions of the cervix[61,62]. To characterize the biochemical composition of the Cervix Chip mucus, structural glycomics analysis of the O- and N-linked glycans of the mucin glycoproteins was performed. O- and N-glycans were first released and then individually profiled using Agilent nanoscale liquid chromatography quadrupole time-of-flight coupled to tandem mass spectrometry (nanoLC-QTOF MS/MS)[61]. The mucus isolated from the epithelial channel of the Cervix Chip contained >50 O- and 350 N-glycans and >11 O- and 129 N-glycans when cultured under continuous and periodic flow, respectively, with the undecorated, sialylated, sialofucosylated, and fucosylated glycan components being the most abundant molecular species identified. Comparison of the O- and N-glycosylation profiles of mucus isolated from Cervix Chips cultured under periodic versus continuous flow revealed that more diverse O- and N-glycan structures were observed in the mucus from the continuous flow chip (Fig. 2a). However, some of the most abundant glycan structures, including sialylated O-glycans (2_2_0_2 │ 28.2), (2_2_0_1 │ 23.5), (3_3_0_2 │ 30.2), (3_3_0_2 │ 31.3) and sialylated (5_4_0_2 │ 27.5) and sialofucosylated (5_4_1_1 │ 25.3) N-glycans, were detected in both chips. Interestingly, higher levels of O-glycans were detected in mucus from the periodic flow chips, whereas an opposite trend was observed for the N-glycans. For example, based on ions counts and area under the curve the sialylated O-glycans (2_2_0_2 │ 28.2) were > 6-fold higher in mucus of periodic flow chip compared to chips exposed to continuous flow, while the sialylated N-glycan (5_4_0_2 │ 27.5) was > 7-fold higher in continuous flow chips (Fig. 2a). In other words, the periodic flow chips preferentially produce a mucus dominated by O-glycans, while the continuous chips predominantly produce N-glycans. When we compared O- and N-glycan profiles of mucus isolated from these Cervix Chips with those produced by cells in Transwell culture, we found they were distinct in that the Transwells failed to produce the different sialofucosylated and fucosylated O- and N-glycans that were abundantly present in mucus produced on-chip under both flow conditions. (Fig. 2a). Importantly, the wide range of mucus O- and N-glycans accumulated on-chip also more closely resembled the heterogeneity of the glycan profiles observed in clinical cervical mucus samples (containing a mixture of endo- and ecto-cervical secretions along with low levels of vaginal secretions) compared to the glycan pattern exhibited by samples collected from Transwell cultures. The Transwell mucus also had high abundance of multiple undecorated O-glycans that were not detected in the chip and clinical mucus samples (Fig. 2a). Mucus from Cervix Chips contained the most commonly found O- and N-glycans present in the clinical human mucus, including >45 % (5 of 11) of the O-glycan and 75 % (9 of 12) of the N-glycan structures (Fig. 2b). These O- and N-glycans exhibited the same structure and configuration (nature, order, and location) of monosaccharide residues, as well as the same

compound retention time, in all sample types. Although most of these common structures were also present in Transwell mucus, the sialofucosylated (2_2_1_2 │ 24.7) O-glycan commonly found in clinical mucus was not detected in mucus from Transwell cultures.

## Modulation of cervical mucus by pH and sex hormone levels

The microenvironment of the lining of the human cervix has a slightly acidic to neutral pH ( ~ 5.6-7) and elevated pH levels are associated with dysbiotic vaginal microflora that can ascend to the cervix and upper female reproductive tract[63]. When we analyzed mucus from Cervix Chips cultured under continuous flow that exhibit features of ecto-cervix which is anatomically closer to the vagina and experiences a more acidic pH ( ~ 3.8-4.5) in vivo, we found that the mucus layer more than doubled in projected area when the cells were cultured at pH 5.4 compared to neutral conditions (pH 7.1) using dark field and fluorescence microscopic side view imaging of WGA-stained cultures (Fig. 3a). Measuring the pH at the inlet and outlet of the apical channel of the chip using a digital microprobe pH meter revealed a significant increase in pH from 5.4 in the chip inflow to 6.2 in the outflow (Supplementary Fig. 3a), which is comparable to clinically reported pH levels in the human cervix in vivo[63]. The acidic pH on-chip did not affect the integrity of the epithelial barrier function or the total amount of Alcian Blue-staining mucus material compared to neutral pH (Supplementary Fig. 3b, c). However, real time qPCR analysis revealed a significant ( > 5-fold) increase in expression of mucin *MUC4* and *SLPI* ( > 2-fold) in the cervical epithelium differentiated under the acidic pH conditions (Fig. 3b), while other major cervical epithelial genes (e.g., *MUC5B*, *ASRGL1*(asparaginase and isoaspartyl peptidase 1)) remained unchanged (Supplementary Fig. 3d). Acidic pH also significantly reduced production of inflammatory TNF-α protein by the cervical epithelium on-chip (Fig. 3c).

Hormonal fluctuations during the menstrual cycle are induced by ovarian endocrine secretions and modulate the cervical microenvironment, including the properties and functions of the cervical mucus[29,30]. Because ovarian tissues were not included in our in vitro cultures, follicular and luteal hormonal phases of the menstrual cycle were simulated in the Cervix Chip exposed to periodic flow (to induce a more endocervical phenotype) by perfusing differentiation medium through the basal channel containing either high (5 nM) β-estradiol (E2) and no progesterone (P4) to mimic the follicular phase or low (0.5 nM) E2 and high (50 nM) P4 hormone levels to recreate the luteal environment[64]. Side view microscopic imaging of the WGA-stained mucus layer showed increased accumulation of mucus on-chip cultured under follicular conditions with high E2 compared to low E2/high P4 luteal conditions (Fig. 3d). Quantification of these results confirmed that the mucus layer was significantly thicker (0.33 vs 0.22 mm) and exhibited a higher level of WGA fluorescence signal intensity (27.1 vs 22.1 gray value) under follicular versus luteal conditions (Fig. 3d, e). The Cervix Chip mucus also exhibited a significantly higher amount of Alcian Blue staining mucus material when exposed to follicular hormone levels (Supplementary Fig. 4a) and this was accompanied by a significantly higher ratio of *MUC5B*: *MUC5AC* gene expression (Supplementary Fig. 4b).

Interestingly, using the clinical Fern Test in which vaginal secretions are allowed to dry on a slide[65], we found that mucus effluents from the follicular and luteal chips formed fern-like and granular patterns respectively, which are similar to the patterns formed by follicular and luteal cervical mucus in clinical studies[66] (Fig. 3f).

In parallel, we carried out glycomic analysis which revealed that the abundances of the sialylated (2_2_0_2 │ 28.2) and disialofucosylated (2_2_1_2 │ 24.7) O-glycans decreased in luteal chip mucus while the monosialylated (2_2_0_1 │ 19.3) O-glycans increased in mucus produced under luteal hormone conditions compared to follicular conditions on-chip (Supplementary Fig. 4c) suggesting a decrease in sialylation. There also was an overall drop in most of the identified sialylated,

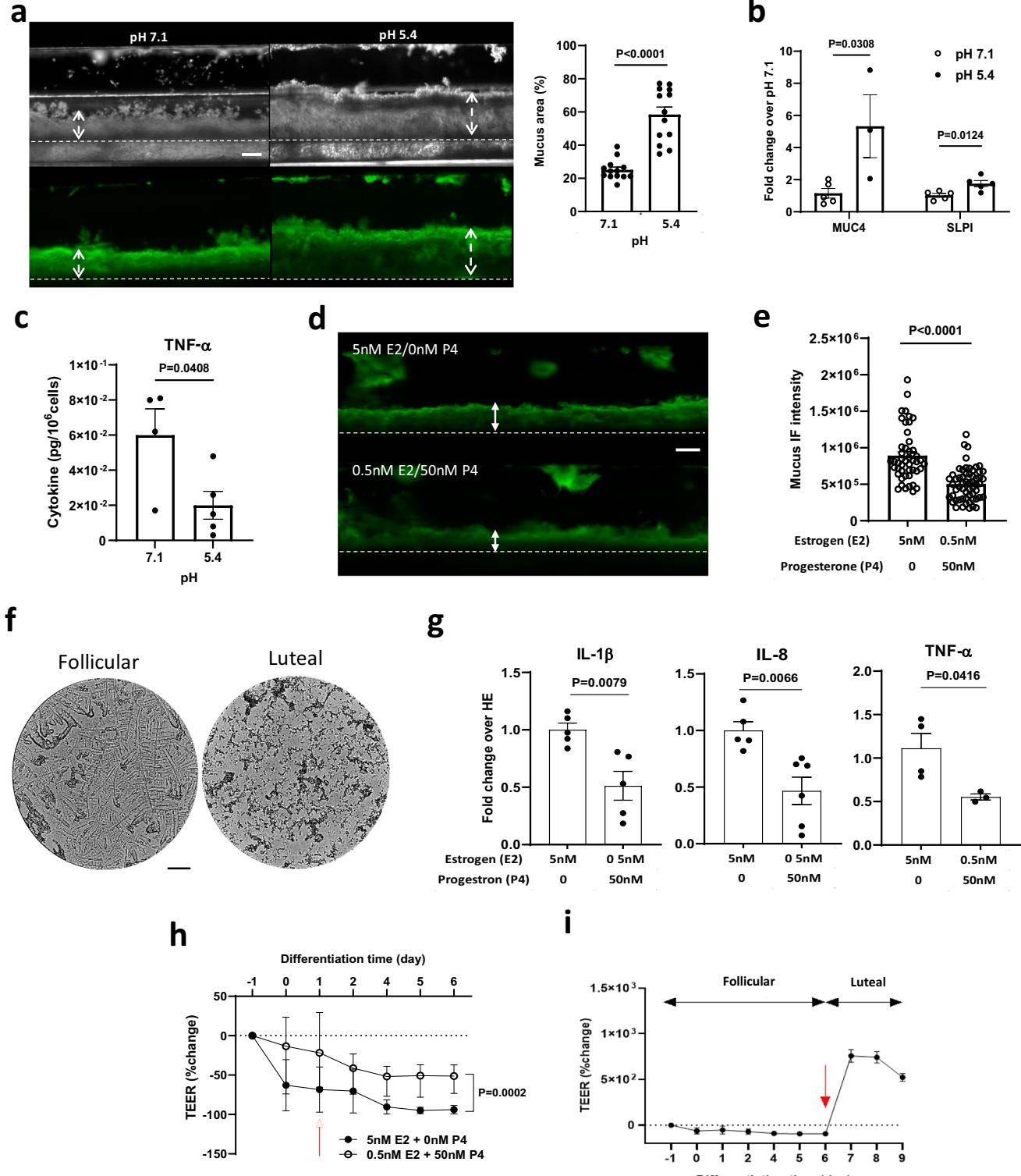

sialofucosylated and fucosylated N-glycans in response to exposure to high progesterone levels on-chip. Importantly, these changes in the mucin glycosylation profile are consistent with reported hormone-dependent mucin glycosylation patterns previously observed in clinical samples[30,62].

The follicular hormone condition with high E2 was also accompanied by significant upregulation of pro-inflammatory cytokines, including IL-1β, IL-8, and TNF-α proteins on-chip compared to the luteal condition (Fig. 3g), which is consistent with the known ability of estrogens to exhibit pro-inflammatory activities[67]. In addition, the

integrity of the cervical epithelial barrier was consistently lower when the chip was cultured under these follicular conditions (Fig. 3h), which again is consistent with the relaxation of the epithelial tight junctions and associated with increased fluid flux across the epithelium leading to higher mucus hydration that has been reported during follicular phase of the cycle[68]. Moreover, when the hormone levels were switched from high estrogen to high progesterone on day 6 of differentiation, the TEER values measured over time in the same cultures using integrated electrodes significantly increased confirming the hormone responsiveness of the cervical epithelial barrier on-chip (Fig. 3i).

**Fig. 3 | Recapitulation of cervical mucus responses to physiological pH and sex hormones in the Cervix Chip. a** (Left) Dark field (top) and fluorescence (bottom) side views of unlabeled and WGA-stained (green) mucus in live Cervix Chips cultured at pH 7.1 compared to pH 5.4. Double headed dashed arrows show mucus thickness (bar, 200 μm). (Right) Graph showing changes in the % of area containing WGA-stained mucus in the apical epithelium channel. **b** Graph showing fold change in expression of cervical epithelial genes encoding mucin 4 (*MUC4*) and secretory leukocyte peptidase inhibitor (*SLPI*) in Cervix Chips cultured at pH 7.1 and pH 5.4. **c** Graph showing TNF-α level in Cervix Chips cultured at pH 5.4 compared to 7.1. **d** Fluorescence side view micrographs of WGA-stained mucus (green) in live cervix chips showing increased mucus layer thickness when cultured with high estrogen (follicular, 5 nM E2 + 0 nM P4) compared to high progesterone (luteal, 0.5 nM E2 + 50 nM P4) levels (double headed arrows show mucus thickness (bar, 200 μm)). **e** Total mucus content (mucus thickness x immunofluorescence intensity)

measured in live Cervix Chips cultured under different hormonal conditions shown in (**d**). **f** Bright field images of the mucus Fern Test collected from Cervix Chips under high estrogen compared to high progesterone hormone conditions (bar, 200 μm). **g** Cytokine secretion levels measured in cervix chips under high estrogen (HE, 5 nM E2 + 0 nM P4) compared to high progesterone (0.5 nM E2 + 50 nM P4) hormone conditions. **h** Changes in tissue barrier function in the Cervix Chip measured using transepithelial electrical resistance (TEER) under high estrogen versus high progesterone hormone levels using the TEER sensor-integrated chip (red arrow indicate start of hormone treatment). **i** Dynamic changes in the epithelial barrier function when hormones are changed from follicular to luteal conditions at day 6 of differentiation (red arrow indicates time that hormone treatment switches from follicular to luteal conditions). Data represents the mean ± s.e.m.; *n* = 13 (**a**) 5 (**b**) 4 (**c**) 6 (**e**) 5 (**g**) and 3 (**h**, **i**) experimental chip replicates for each group. Source data and statistical tests are provided as a Source Data file.

## Colonization of the Cervix Chip by optimal and non-optimal commensal bacterial communities

*L. crispatus* dominant microbiota are known to be associated with optimal, healthy human cervico-vagina mucosa in vivo[44,69], while *G. vaginalis* is commonly found in dysbiotic bacterial communities associated with BV[70]. Commensal bacterial communities are present in the cervical canal, uterus, and fallopian tubes as well as in the vagina[71], however, transmission of pathogenic bacteria from the vaginal canal to the upper reproductive organs through the endocervix is a major cause of prenatal labor and infant mortality. To model the impact of healthy and dysbiotic microbiome, we therefore cultured a bacterial consortium containing three optimal *L. crispatus* strains (C0059E1, C0124A1, C0175A1)[72] in Cervix Chips cultured under periodic flow to mimic the effect of vaginal microbiome on the endocervix. The follicular phase of the menstrual cycle with higher level of estradiol has shown to strongly associate with the presence of a healthy state and a more stable *L. crispatus* dominated microbiome in the female reproductive tract[28]. To understand how dysbiotic bacteria initially invade and destruct the host defense mechanism in the healthy state, we therefore simulated healthy chips and dysbiotic endocervix by respectively culturing either a consortia of three *L. crispatus* or two non-optimal *G. vaginalis* strains under high estrogen conditions (5 nM E2) in these chips.

Phase contrast and immunofluorescence microscopic imaging of differentiated Cervix Chips colonized with *L. crispatus* for 3 days confirmed that the epithelial cell layer remained intact and cells appeared refractile even though we could detect the presence of bacteria closely associated with MUC5B-containing mucus on the surface of the epithelium (Fig. 4a). In contrast, when the *G. vaginalis* consortium was cultured on-chip under similar conditions, we observed a greater abundance of *G. vaginalis* bacteria and this was accompanied by disruption of the epithelial cell layer, cell detachment, and appearance of smaller pyknotic nuclei (Fig. 4b).

The Cervix Chip maintained a stable engrafted population of both *L. crispatus* and *G. vaginalis* bacteria throughout the 3 days of co-culture as determined by quantifying the total number of live culturable bacteria collected every 24 hrs from the apical channel effluents and from epithelial tissue digests at the end of the experiment compared to the initial inoculum (Fig. 4c). This method also confirmed that co-culture of *L. crispatus* bacteria with the epithelium did not affect epithelial cell number whereas colonization with *G. vaginalis* produced a significant loss of cervical epithelial cells compared to control non-inoculated and *L. crispatus* colonized chips (Fig. 4d). In contrast, measurement of pH levels in the epithelial channel after 72 h of co-culture with both *L. crispatus* and *G. vaginalis* showed comparable levels to those in control chips with no bacteria (Supplementary Fig. 5a). However, while *L. crispatus* colonization did not affect the epithelial barrier integrity, *G. vaginalis* infection significantly compromised the epithelial barrier at 72 hrs as indicated by an increase in $P_{app}$ when compared to the chips colonized with *L. crispatus* consortia

or control chips without bacteria (Fig. 4e). In addition, the presence of *G. vaginalis* consortia significantly upregulated production of multiple pro-inflammatory cytokines, including IL-1α, IL-1β, IL-6, IL-8, and TNF-α in the upper channel effluents of the chip, while *L. crispatus* consortia downregulated this inflammatory response compared to the non-inoculated control chip when measured after 3 days of co-culture (Fig. 4f).

We further explored cervical epithelial cell molecular responses to healthy versus dysbiotic consortia using proteomic analysis of the cervical epithelial cells on-chip after 72 h culture with and without commensal (*L. crispatus*) versus dysbiotic (*G. vaginalis*) bacteria using liquid chromatography-tandem mass spectrometry (LC-MS/MS). These results showed that compared to control chips with no bacteria, colonization with *G. vaginalis* upregulates expression of several proteins in the cervical epithelial cells that are involved in the cell processes associated with the pathogenesis phenotype including HLA-C (Class 1 HLA antigens) and TRIM25 (an E3 ubiquitin ligase enzyme) proteins regulating adaptive and innate immune responses, free ribosomal proteins (RPL15 and RPL30) participating in cellular development and cell cycling associated with cell apoptosis and tumor pathogenesis[73,74], ACADVL (very long-chain specific acyl-CoA dehydrogenase) protein that compromises tissue integrity and function through catalyzed mitochondrial beta-oxidation[75], cells membrane vesicular trafficking protein (EXOC8) exploited by microbial pathogens for cell entry and infection[76,77], and enzyme transglutaminase 1 (TGM1) found in the epithelium outer layer which may be involved in cell shedding[78] as part of the host defense mechanism to *G. vaginalis* (Fig. 4g).

Interestingly, colonization with *L. crispatus* significantly increased CSRP1 (cysteine and glycine-rich protein 1), 9PTGFRN (prostaglandin F2 receptor negative regulator), and CSTB proteins, while down regulating secretion of a free ribosomal protein (PRS28) in the cervical epithelial cells compared to no bacteria condition (Fig. 4g). The proteins that increased are all highly expressed in the female reproductive tissues[79] and are known to be involved in important cellular regulatory processes including cell development, differentiation, and protection against proteolysis enzymes[80,81]. These data confirm the important role that commensal *L. crispatus* bacteria play in regulating host cervical epithelial cell functions and responses as is observed in healthy cervix in vivo.

We also identified proteins that are differentially expressed in *G. vaginalis* compared to *L. crispatus* colonized Cervix Chips and found that the CSRP1 and PTGFRN proteins are significantly downregulated under these dysbiotic conditions. Furthermore, TRM25, RPL30, S100A7 (psoriasin), and ANKRD22 (ankyrin repeat domain-containing protein 22) proteins that are aberrantly expressed in cervical and ovarian cancers[82–84] and that induce or are induced by different proinflammatory cytokines and chemokines[85] were found to be significantly upregulated in dysbiotic compared to healthy (*L. crispatus*-containing) Cervix Chips (Fig. 4g).

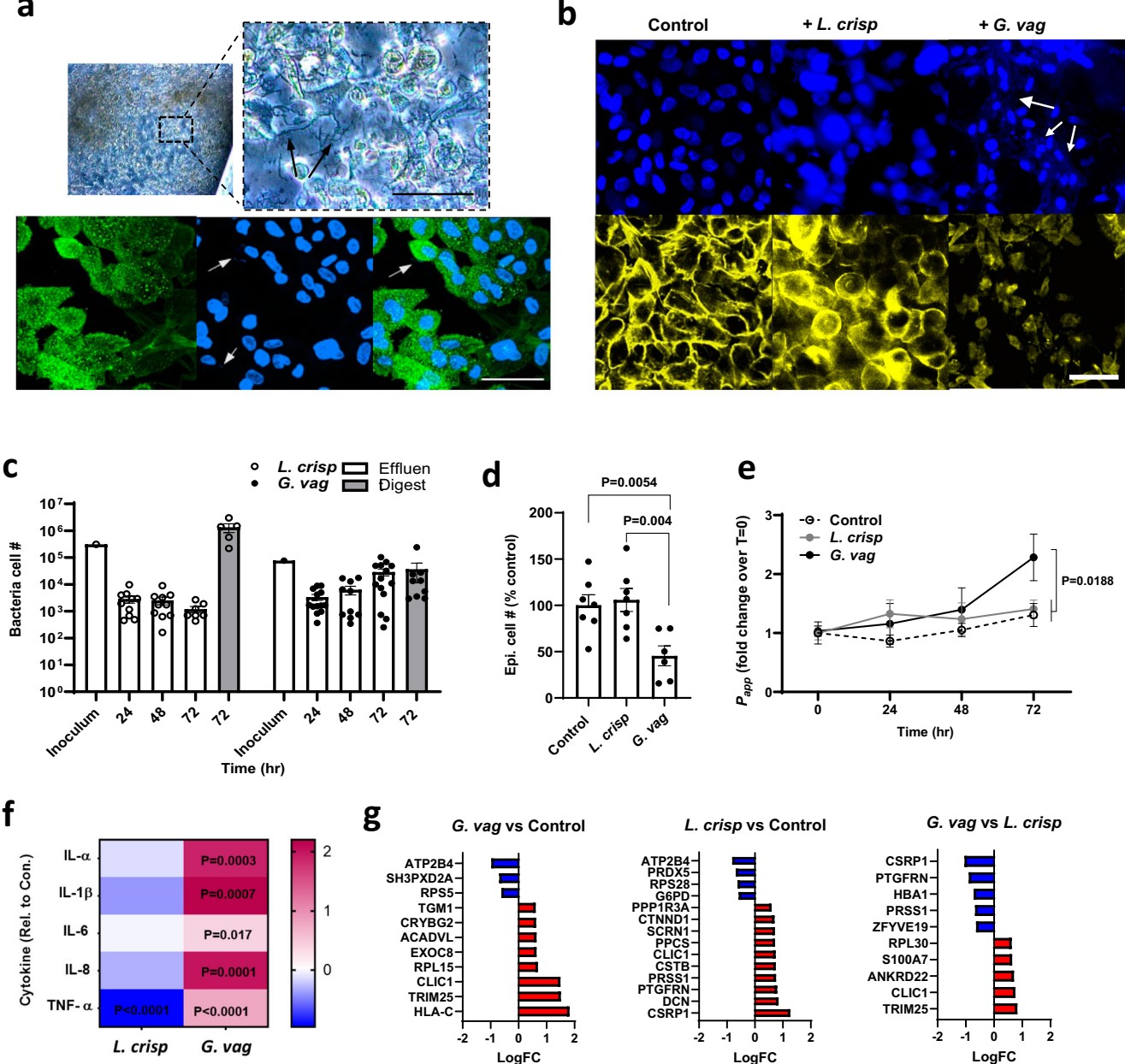

**Fig. 4 | Modeling cervical epithelial host interactions with *L. crispatus* and *G. vaginalis* consortia in Cervix Chips. a** Phase-contrast (Top) and immuno-fluorescence (bottom) microscopic views from above of the Cervix Chip co-cultured with *L. crispatus* consortia. (Top) Phase-contrast microscopic view (left) and higher magnification image (right) of live chip cervical epithelium colonized with *L. crispatus* bacteria. (Bottom) Immunofluorescence micrographs of cervical epithelium stained for MUC5B (green) (left), nuclei with Hoechst (blue) (middle) and the overlay image (right) (Black and white arrows show *L. crispatus* bacteria colonized on-chip (bar, 50 µm)). **b** Immunofluorescence micrographs of cervical epithelium stained for F-actin (yellow) and nuclei with Hoechst (blue) in Cervix Chip co-cultured with no bacteria (left), *L. crispatus* (middle) and *G. vaginalis* (right) (White arrows show *G. vaginalis* bacteria colonized on the epithelium (bar, 50 µm)). **c** Enumeration of the total non-adherent (Effluent; white bar) and adherent (Digest; gray bar) bacteria in the Cervix Chip during and at the end of co-culture, respectively, with the *L. crispatus* or *G. vaginalis* consortia compared to the initial inoculum in the Cervix Chip. **d** Graph showing percentage change in the viability of

cervical epithelial cells after 72 h of co-culture with *L. crispatus* or *G. vaginalis* consortia in the Cervix Chip compared to control chip without bacteria. **e** Graph showing fold change in the cervix Chip barrier permeability during co-culture time with *L. crispatus* or *G. vaginalis* consortia compared to non-inoculated, control chip as measured by apparent permeability ($P_{app}$). **f** Heat map showing Cervix Chip epithelium innate immune response to *L. crispatus* or *G. vaginalis* consortia at 72 h post inoculation quantified by levels of IL-1α, IL1-β, IL-6, IL-8, and TNF-α, in the epithelial channel effluents. The color-coded scale represents LOG10 fold change in cytokine levels over control chip. **g** Proteomic analysis showing significantly up- (red) and down- (blue) regulated proteins in the cervical epithelial cells on-chip in response to *G. vaginalis* and *L. crispatus* consortia as compared to control chips without bacteria, as well as to each other. $P ≤ 0.05$, log2FC ≥ |0.58| for all the pre-sented protein gene symbols. Data represent the mean ± s.e.m.; $n = 10$ (**c**) 6 (**d**) 10 (**e**) 10 (**f**) and 3 (**g**) experimental chip replicates for each group. Micrographs in (**a**, **b**) are representatives of three separate experiments. Source data and statistical tests are provided as a Source Data file.

## Modulation of the Cervix Chip mucus by the microbiome

Colonization of the Cervix Chip with *L. crispatus* bacteria resulted in a significant increase in mucus layer thickness and WGA-stained fluorescent signal intensity along the entire length of the chip compared to chips co-cultured with *G. vaginalis* or without any bacteria as visualized on day 3 of co-culture (Fig. 5a-c), although there was no significant change in the total mucin content of the effluents from the epithelial channel (Supplementary Fig. 5b). Interestingly, Fern Testing of the mucus collected from the chip effluents revealed that colonization with *L. crispatus* increased the length and branching degrees of the mucus ferns (up to quaternary) with more observed crystallization compared to mucus from chips without bacteria, while *G. vaginalis* reduced these features in a manner similar to that previously observed in clinical samples of atypical mucus[86] (Fig. 5d). The presence of *G. vaginalis* bacteria also significantly reduced the relative abundance of major sialylated O-glycans (e.g., 2_2_0_2 | 28.9 structure) and this was accompanied by a significant increase in the abundance of multiple undecorated O-glycans (Fig. 5e, f). Importantly, similar changes in sialylated O-glycans have been previously observed in BV-associated mucus in vivo[30,87]. The loss of sialic acids in the mucus may reduce interactions between mucin molecules and contribute to the decreased fern branching that we observed in the Cervix Chip mucus.

## Discussion

For decades, development of new treatments for dysbiosis, BV, and other reproductive health conditions has been limited by our poor understanding of the complex host-microbiome interactions and the role of cervical mucus in host immunity of the lower reproductive tract. It also has been impossible to tease out contributions from mucus produced by the endo-cervix versus ecto-cervix. Advancements have been held back by the lack of in vitro models that faithfully recapitulate the physiology and pathophysiology of the human cervix microenvironments[2,3,10,44]. Here we described development of preclinical human Cervix Chip models that recapitulate the physiology of the endo- and ecto-cervical mucosa as well as their responses to microenvironmental, hormonal, and microbiome influences. The dual-channel design of the microfluidic chip enabled formation of a human cervical mucosa in vitro containing stromal cells interfaced with cervical epithelium, which is critical given the known importance of the stroma for control of cervical epithelial growth, function, and hormone-stimulated secretions.[10,22,88,89]. Indeed, under these conditions we observed formation of a mucus-producing cervical epithelium with a tight permeability barrier that recapitulated many features of the cervical mucosa observed in vivo.

Human cervix is not perceived to experience a significant form of fluid flow similar to that in the intestine or arteries, however, it continuously produces mucus, hydrated by interstitial fluids, and clears it from the surface of the epithelium as it moves (flows) to the vagina by mucociliary transport. This flow can be further augmented by gravity and changes in the endocrine hormones and microenvironmental factors in the reproductive tract. Thus, mucus flow, even if minimal, can influence epithelial cell function through associated application of fluid shear stresses or other physical forces (e.g., osmotic) as it does in many other organs or by preventing mucus over-accumulation and hence, it is important that this be at least explored as a contributing factor in any in vitro model. The Organ Chip culture system enabled us to explore two different flow conditions, including a gentle continuous flow and a less frequent periodic flow. Interestingly, we discovered that these dynamic flow conditions provide biomechanical cues that significantly influence cervical epithelial cell differentiation as well as mucus composition and innate immune responses. Specifically, we found that periodic flow promoted endo-cervical differentiation on chip while continuous flow induced expression of more ecto-cervical phenotype. These findings are consistent with the profound effects that dynamic fluid flow and other biomechanical cues have been

shown to have on differentiation of other types of epithelial cells (e.g., intestine, kidney, etc.)[40,90,91]; however, to our knowledge this has not been studied to date in the context of cervical development, and only the effect of host-induced biomolecular signals (e.g., Wnt) has been previously shown to induce differentiation of distinct endo- and ecto-cervical epithelial lineages[92]. Interestingly, we also found that while periodic flow conditions that included intermittent static periods produced effects that promoted endo-cervical specialization which were more similar to those observed when the same epithelial and stromal cells were cultured in the same medium in static Transwell inserts, there were also distinct differences, including differences in epithelial barrier integrity, mucus composition, and collagen expression profile with the chip more closely resembling features observed in vivo.

Mucus secretion is the primary function of the cervix in the female reproductive tract that protects the epithelium against pathogen invasion. We showed here that the engineered Cervix Chips accumulate thick mucus layers that are amenable to real-time quantitative analysis in terms of their thickness and chemical composition. This approach revealed that the Cervix Chips respond physiologically to both hormonal and bacterial influences in a manner similar to that observed in vivo. The abundance of the mucus produced on-chip allowed for non-invasive longitudinal analysis of its formation as well as collection of both soluble mucus materials that flow out in the effluent of the epithelial channel and adherent mucus that remains adherent to the epithelium when the cultures are sacrificed at the end of the study. This is a major advantage compared to existing in vitro and in vivo models used to study cervix physiology and pathophysiology. By using image-based, biochemical, biomolecular, transcriptomic, and glycomic analyses, we were able to demonstrate that the mucus produced in the Cervix Chips faithfully recapitulated multiple features of human cervical mucus produced in vivo. In particular, high resolution glycomic analysis was used to quantify and characterize cervical mucus glycan subtypes and molecular structures, which also confirmed that the mucus produced on-chip under dynamic flow conditions more closely resembled clinical mucus isolated from cervical samples than that produced by the same cervical cells in static 2D cultures or Transwell.

Clinical samples of vaginal secretions contain a mixture of ecto- and endo-cervical mucus as well as a lower level of mucus produced by the vaginal epithelium. Thus, little is known about how each of these regions contribute to the composition of cervico-vaginal mucus. In contrast, our in vitro Cervix Chip enabled glycomic analysis of mucosal secretions from Cervix Chips that have transcriptomic signatures similar to either endocervix or ectocervix. These studies revealed the different abundance of O- and N-glycosylated mucins in these two different epithelial phenotypes. Teasing out of the unique contribution of each cervical sub-lineages is important for understanding the physiology and pathophysiology of the lower reproductive tract in diseased or non-functional organs such as cervical cancer and fibrosis that can affect different parts of the cervix. Thus, these findings provide a baseline for future studies analyzing how the hormonal, bacterial, or host-derived modulations of mucus glycosylation structure can influence cervico-vaginal physiology and pathophysiology, while much has been previously reported in studies on intestinal mucus-microbiome interactions[93–95]. Importantly, one major advantage of the Organ Chip model is that it allows many potential contributing factors to be varied independently or in combination, which is impossible in animal models or in clinical studies where the effects of mucus cannot be separated from those induced by external bacterial or microenvironmental factors, nor can contributions of endo-cervical mucus be separated from those produced by ecto-cervical secretions.

Microenvironmental factors, such as pH and hormone fluctuations during menstrual cycle, pregnancy, and menopause, also can modulate intracellular functions of the mammalian cells, mucus

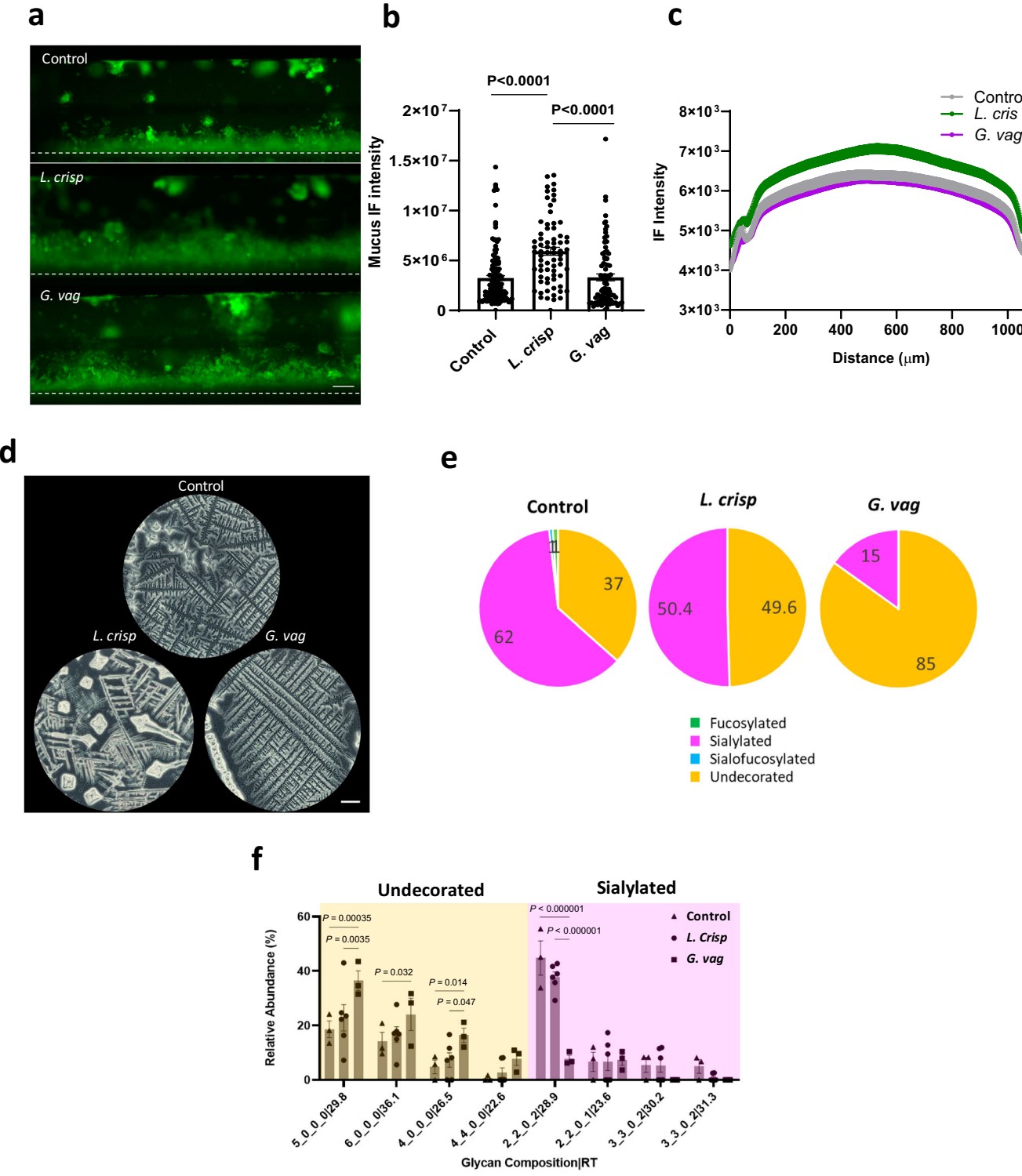

**Fig. 5 | Modeling modulation of cervical mucus with *L. crispatus* and *G. vaginalis* consortia on Cervix Chip. a** Fluorescence side view micrographs of WGA-stained mucus (green) in live Cervix Chips co-cultured with *L. crispatus* (middle) or *G. vaginalis* (bottom) compared to control chip (top) without bacteria (white dashed line indicates porous membrane) (bar, 200 μm). **b** Total mucus content (mucus thickness x immunofluorescence intensity) measured in live Cervix Chips cultured with *L. crispatus* or *G. vaginalis* compared to non-inoculated control chip. **c** Graph showing the spatial distribution of the live mucus immunofluorescence intensity along the width of the epithelial channel in the Cervix Chip co-cultured with *L. crispatus* or *G. vaginalis* consortia compared to control chip. **d** Bright field images of the mucus Fern Test collected from Cervix Chips co-cultured with *L. crispatus* or *G. vaginalis* consortia compared to control chip without bacteria (bar,

200 μm). **e** Pie charts representing the relative abundances of O-glycan types in the Cervix Chip mucus after 72 h of co-culture with *L. crispatus* or *G. vaginalis* consortia compared to control chip without bacteria quantified using nanoscale liquid chromatography quadrupole time-of-flight coupled to tandem mass spectrometry (nanoLC-QTOF MS/MS). **f** Relative abundance of undecorated and sialylated O-glycans in the Cervix Chip mucus collected after 72 h of co-culture with *L. crispatus* or *G. vaginalis* consortia compared to control chip. Data represent the mean ± s.e.m.; *n* = 120 fields of view from 3 (**b**) and 6 (**f**) experimental chip replicates for each group. Micrographs in (**a**) are representatives of six separate experiments for each group. Source data and statistical tests are provided as a Source Data file.

properties, and microbial communities, as well as host-microbiome homeostasis[18,96]. Teasing out the contribution of each of these factors in the cervico-vagina-microbiome homeostasis has been almost impossible due to their simultaneous occurrence in vivo. Using Cervix Chips, we demonstrated that each of these contributing factors can be independently and physiologically modeled and studied in vitro. Importantly, using this more well defined characterization platform, we showed that physiological responses obtained on-chip are similar to those observed in vivo[43,97].

With the increasing recognition of the importance of the microbiome for health and disease of the female reproductive and genital tracts[69,71], it is critical to develop in vitro models that can be used to study the complex host-microbiome interactions. We leveraged the Cervix Chips to model healthy- and dysbiotic-associated cervical microenvironments in vitro using clinically derived isolates of *L. crispatus* (optimal) and *G. vaginalis* (non-optimal) bacterial communities. Colonization of the Cervix Chips resulted in producing of the cervical mucus discharge that closely resembled samples obtained clinically, including similar results in Fern Tests that are commonly used in clinic as diagnostic. Co-culture of Cervix Chips with optimal *L. crispatus* bacteria exhibited a stable epithelial layer, good barrier function, a quiescent immune state, and mucus production with clinically relevant composition and secretion of cellular proteins similar to those produced by healthy cervix in vivo. In contrast, when these chips were co-cultured with disease-associated *G. vaginalis* bacteria, we observed damage of the cervical epithelium, cell death, compromised barrier, and altered mucus properties in addition to an enhanced inflammatory response and secretion of proteins that have been previously associated with pathological conditions, which are all major phenotypic signatures of a dysbiotic cervix in vivo[11,69,98]. Although colonization of the lower reproductive tract by *L. crispatus* bacteria has been associated with a healthy cervicovagina microenvironment, its positive influence on host cell biology and function is not fully understood. The Cervix Chip enabled studying of changes in the cervical cell proteome in response to healthy and dysbiotic bacteria, which confirmed the important role of *L. crispatus* in inducing host cellular responses that promote normal healthy tissue development and defense mechanisms[79].

On a biochemical level, we found that *G. vaginalis* significantly reduces the abundance of sialylated O-glycans in the Cervix Chip mucus. Dysbiotic bacteria including *G. vaginalis, Prevotella, and Bacteroides* express sialidases enzymes[99] that cleave the sialic acid shielding cap of the mucin glycans and making the mucin polymers more vulnerable to further degradation by other dysbiotic bacteria proteases and mucinases enzymes[30,100,101]. This can result in profound effects on the mucus layer, including reduced molecular interactions between the mucin glycoproteins and easier shedding of the mucosal layer from the epithelial surface. The effect of *L. crispatus* on the cervical mucus and its glycosylation profile is, however, not fully understood, aside from its known positive effect in protection against pathogens[102]. We showed that colonization of the Cervix Chip with *L. crispatus* maintains a high level of sialylated O-glycans, essential for mucosal protection against dysbiotic bacteria and pathogens aggression[103]. In addition, this study revealed that these optimal *L. crispatus* communities may protect against pathogenic bacteria by inducing formation of a thicker mucosal layer and enhancing innate immunity.

The primary focus of this work was to model human cervical mucosal physiology and its impact on host-microbiome interactions in vitro, however, there are some caveats to this model that can be improved in the future. The commercially available cervical epithelial cells we used to benchmark against past in vitro models contain a mixed population of endo- and ecto-cervical cells, which are not ideal for development of fully specialized endo- and ecto-Cervix Chips. Although their use in our Cervix Chip model revealed the effect of

different flow environments on preferential specialization of endo- and ecto-cervical epithelial cell lineages, the model would be strengthened by isolating primary epithelial and stromal cells from each of these distinct regions of the human cervix in the future. The complexity of the human cervix biology and immunobiology stems from dynamic molecular signaling among multiple cell types including the epithelial, stromal, endothelial, immune cells, and the bacterial communities in the lower reproductive tract[104–108]. Thus, while we replicated many important physical and chemical features and biology of the cervical mucosa in our Cervix Chips lined by only epithelium and stromal fibroblasts, it will also be important to integrate endothelial cells and immune cells into these chips as they are important contributors to the tissue immune responses and pathophysiology. The ability to add increasing complexity by integrating additional cell types as desired, including endothelium and immune cells, is a major advantage of the Organ Chip technology[109]. Finally, when an infection ascends from the vagina to the cervix and upper reproductive organs, vaginal secretions and vaginal microbiome metabolites also pass to the cervix along with the dysbiotic bacteria. As this can contribute to cervix pathophysiology under these dysbiotic conditions, it would be interesting to model these conditions by fluidically linking the Cervix Chip with the Vagina Chip in future studies[42].

Taken together, the Cervix Chip models provide experimental platforms for study of human cervical physiology and pathophysiology that far surpass the previously reported in vitro models in terms of their ability to recapitulate endo- and ecto-cervical epithelial differentiation, barrier formation, mucus production, and functional responses to hormonal, microenvironmental and bacterial stimuli. While in vitro models of the cervix have been used previously to study host responses to healthy- and diseased-associated bacteria and microbicides[35,110] they commonly used immortalized or cancer epithelial cell lines, lacked an in vivo like cervical epithelial-stromal interface, and failed to demonstrate clinically relevant mucus production. Taken together, these findings suggest that these microfluidic human Cervix Chips that offer many advantages over conventional culture systems may provide more reliable preclinical models for analysis of host-microbiome interactions as well as test beds for biotherapeutic treatments and interventions that aim to treat BV and improve reproductive health among millions of women around the world.

## Methods

All methods were carried out in accordance with the approval of the Institutional Review Board of Wyss Institute for Biologically Inspired Engineering at Harvard University (Protocol number IRB22632) and Mass General Brigham (Protocol number 2015P001859). When human research participants were involved, informed consent was not obtained because the samples were deidentified. No compensation was provided to the participants.

### Cervix Chip seeding and culture

A microfluidic organ chip (Emulate, CHIP-S1™) made of transparent, flexible polydimethylsiloxane (PDMS), and constitute of two parallel microchannels (1 mm wide × 1 mm high apical channel and a 1 mm wide × 0.2 mm high basal channel) separated by a porous membrane (50 μm thickness, 7 μm pores with 40 μm spacing) similar to a previously published design[111] was used to create the Cervix Chip model. Before cell seeding, the polymeric surfaces of the membrane were chemically functionalized with 0.5 mg/ml ER1 in ER2 buffer (Emulate™ Inc., USA) polymerized under an ultraviolet lamp (Nailstar, NS-01-US) for 15 min following by sequential washing with ER2 buffer and PBS. The PDMS membrane was coated with 500 μg/ml collagen IV (Sigma-Aldrich, Cat. no. C7521) in the apical channel and with 200 μg/ml Collagen I (Advanced BioMatrix, Cat. no. 5005) and 30 μg/ml fibronectin (Corning, Cat. no. 356008) in DPBS in the basal channel

overnight at 37 °C. The channels were washed with DPBS and culture medium before seeding cells.

Primary cervical epithelial (CE) cells (LifeLine Cell Technology Cat# FC-0080, Donors ID: 6225, 6226, 6227) that contain a mixture of ecto- and endo-cervical epithelial cells were expanded in cervical expansion medium composed of a cervical basal medium (LifeLine Cell Technology, LM-0055) supplemented with cervical growth factors (LifeLine Cell Technology, SKU:LL-0072) and 50 U/ml Penicillin-Streptomycin (Gibco™, 15070063). Primary cervical fibroblast cells were isolated from healthy cervical tissue, obtained from hyster-ectomy procedure on a 37 year-old female (at-birth assigned gender), using a previously described method[112]. Briefly, cervical tissue was enzymatically digested in 1.25U/ml Dispase (Worthington Biochemical Corporation, Cat. no. LS02109) and 0.4 mg/mL collagenase V (Sigma, C-9263) solution in RPMI 1640 medium (ThermoFisher Scientific, 21875-034) containing 10% FBS with gentle shaking at 37 °C for 30–60 min. The digestion supernatant was passed through a 100 μm cell sieve (Corning, 431752) and the flow-through was collected for initial stromal cell culture in Advanced DMEM/F12 (ThermoFisher Scientific, 12634010) containing 10 %FBS and Penicillin-Streptomycin (GibcoTM, 15070063). The cells were subsequently expanded in fibroblast expansion medium; fibroblast basal medium (ATCC® PCS-201-030™) supplemented with fibroblast growth factors (ATCC® PCS-201-041™) and 50 U/ml Penicillin-Streptomycin (Gibco™, 15070063).

To create the Cervix Chips, primary cervical fibroblasts (P5, $0.65 \times 10^6$ cells/ml) were first seeded on the basal side of the porous membrane by inverting the chip for 2 h in fibroblast cell growth medium to allow attachment of the cells to the porous membrane, followed by flipping the chips again and seeding the primary cervical epithelial cells (P5, $1.5 \times 10^6$ cells/ml) on the apical side for 3 h in epi-thelial cell growth medium. The respective media of the chips were refreshed for each channel and the chips were incubated at 37 °C, 5 % $CO_2$ under static conditions overnight. The chips were then connected to the culture module instrument (ZOË™ CULTURE MODULE, Emulate Inc., USA) to enable controlled perfusion of medium in the chips using pressure driven flow. The Cervix Chips were cultured using a periodic flow regimen in which cervical growth medium was flowed through the apical channel for 4 h per day at 30 μl/hr with continuous flow of fibroblast growth medium basally at 40 μl/hr or using a continuous flow regimen that involved continuous perfusion of both apical and basal channels with the same respective media but at 40 μl/hr. We chose these flow rates because they produce a low level of shear stress (0.0003 dyne/cm² in the apical channel and 0.012 dyne/cm² in the basal channel) that was previously found to support effective epithelial differentiation in a human Vagina Chip[42]. After 5 days the apical med-ium was replaced by Hank's Buffer Saline Solution (HBSS) (Thermo Fisher, 14025076) while being fed through the basal channel by con-stant flow of cervical epithelial medium (LifeLine Cell Technology, Cat. no. LL-0072) supplemented with 5 nM female sex hormone estradiol-17β (E2) (Sigma, Cat. no. E2257) to promote epithelial differentiation and 50 μg/mL Ascorbic acid (ATCC, Cat. no. PCS-201-040) to support stromal fibroblast growth and increase collagen synthesis[113] in con-tinuous regimen cultures. Under both flow regimens, the apical med-ium was replaced at day 2 of differentiation by a customized HBSS with low buffering salts and no glucose (HBSS (LB/-G)) with pH ~ 5.4 to better mimic the acidic microenvironment of the cervix in vivo and then the chips were cultured for 5 additional days (7 total days under differentiation conditions; 12 days of culture on-chip). Functional assessment of hormone levels after 7 days of continuous perfusion of the chips confirmed that the levels remained constant and thus, the absorbance saturation equilibrium between any potential absorption of the hormones into the PDMS and the perfusing medium was established[114]. Chip cultures were maintained in an incubator con-taining 5% $CO_2$ and 16–18% $O_2$ at 85–95% humidity. The pH of the chip inflows and effluents for apical and basal channels were measured using SevenGoDuo SG68-B pH meter with InLab Ultra-Micro-ISM probe (METTLER TOLEDO, USA).

## Transwell culture

Transwell® cultures were created with the cervical epithelial and stroma cells using 6.5 mm diameter inserts with 0.4 μm pore size polyester membrane (Corning, Inc.). The surfaces of the porous membrane were coated with collagen IV in the apical side and Collagen I and fibronectin on the basal side following the same method described for the Chip culture. The cell seeding density was adjusted for the TW inserts due to their larger surface area compared to the chip to allow for a comparable cell seeding density in both culture systems. Primary cervical fibroblasts (P5, $0.05 \times 10^6$ cells/ml) were first seeded on the basal side of the porous membrane by inverting the insert and 2 h incubation in fibroblast cell growth medium, followed by flipping the Transwells in the well plate and seeding the primary cervical epi-thelial cells (P5, $0.5 \, 10^6$ cells/ml) on the apical side for 3 h in epithelial cell growth medium. The medium in each chamber of the Transwell was refreshed with respective medium and the inserts were incubated at 37 °C, 5% $CO_2$ under static conditions. The media of the Transwell inserts were changed every day throughout the culture time following the same media composition change regimen described for the chip culture.

## Immunofluorescence microscopy

Cell cultures or Cervix Chips were first washed with DPBS in the apical and basal channels, fixed with 4% paraformaldehyde (FisherScientific, Cat. no. 50-980-487) for 20 min before being washed with DPBS and stored at 4 °C. Before immunostaining, the chips and cells were per-meabilized with 0.05 % triton X-100 (Sigma) in Tris-buffered saline (TBS) for 10 min at room temperature, then blocked in 5 % normal donkey serum (Sigma-Aldrich, D9663), for 1 h at room temperature, followed by overnight incubation in primary-antibodies diluted in 5 % donkey serum at 4° C on a rocking plate. The samples were then incubated with corresponding secondary-antibody for 1 h at room temperature; nuclei were stained with Hoescht 33342 (Life Technolo-gies, H3570, 1:1000) for 10 min at room temperature after secondary-antibody staining. Fluorescence imaging was performed using a con-focal laser-scanning microscope (Leica SP5 X MP DMI-6000) following by image processing using Imaris software (version V9.3). Primary antibodies used in these studies included ones directed against MUC5B (Abcam, Cat. no. ab87376, 1:200), KI67 (Thermo Scientific, Cat. no. RM-9106-S, 1:200), cytokeratin 18 (Abcam, Cat. no. ab668, 1:500), cytokeratin 7 (Abcam, Cat. no. ab209601), cytokeratin 14 (Abcam, Cat. no. ab51054), F-actin (Abcam, Cat. no. ab176757), vimentin (Abcam, Cat. no. 195878, 1:100), estrogen receptor (Abcam, Cat. no. ab32063, 1:200), and progesterone receptor (Abcam, Cat. no. ab2765, 1:200). Secondary antibodies included Donkey anti-rabbit Alexa Flour 647 (Jackson lab, Cat. no. 715-605-151 (1:500) and 715-605-152 (1:500)) and Donkey anti-rabbit Alexa Flour 488 (ThermoFisher, Cat. no. A21206, 1:500). For pseudo-H&E staining, the Cervix Chips were formaldehyde fixed, washed, frozen, sectioned using a cryotome, and sections were stained with Eosin Y solution (Abcam, cat. no. ab246824, used without dilution) for 30 seconds at room temperature followed by 15 min staining with 1 μg/mL 4',6-diamidino-2-phenylindole (DAPI; Invitrogen, cat. no. D1306, dilution 1:1000) before microscopic imaging. Second harmonic generation microscopy of the Cervix Chip was carried out to image collagen fibers in the stroma as previously described[53]. Briefly, the chip was placed on a cover glass (VWR micro cover glass, 24 × 50 mm Cat. no. 48382 136) and imaged using a 25 x water objec-tive using a confocal laser-scanning microscope (Leica SP5 X MP DMI-6000). The imaging plane was chosen to correspond to the fibroblast cell layer adjacent to the chip porous membrane. Multiphoton (MP) laser was tuned to 900 nm and Second Harmonic images were col-lected using a 438–480 nm bandpass filter.

Validation data for all the described antibodies are available on the manufacturer's website, which were verified with the provided Data Sheets from the manufacturers. We also confirmed the specificity of the antibodies for human immunofluorescence staining by validating and optimizing the antibodies dilutions with human cervical tissue samples.

## Epithelial barrier function

Real-time trans-epithelial electrical resistance (TEER) was measured in Cervix Chips using an in-house TEER-sensor integrated Organ Chip device developed based on our previous TEER-integrated OOC system[55]. Briefly, the TEER-integrated sensor chips constitute of dual microfluidic channels with the design dimensions described earlier. Gold electrodes patterned on polycarbonate substrate were integrated into the chips following a layer-by-layer assembly approach that places three electrodes on each side of the cervical epithelium-stroma layer. The TEER-sensor chip was mounted on a printed circuit board (PCB) and assembled into a standalone Organ Chip unit that was multiplexed and directly connected to an external potentiostat, which enabled real-time and remote TEER measurements of the Cervix Chips from outside the incubator throughout the culture time. TEER measurements were performed by applying an electrical current (10 µA) to the chip through one set of electrodes at a frequency sweep of 10 – 10000 Hz and measuring the drop in the potential by another set of electrodes in the chip using a CompactStat.h Mobile electrochemical interface (B32121, Ivium Technologies B. V., The Netherlands). Four-point impedance measurements were taken periodically throughout the culture time using IviumSoft (V4.97, Informer Technologies, Inc.). The cervical epithelium TEER was presented as corrected TEER obtained from measured impedance at 100 Hz subtracted from that measured at 100,000 Hz, which represent the chip media resistance. At <100 Hz frequency the impedance signal is dominated by the TEER[55]. TEER of the cervical epithelial cells on Transwell was measured using the EVOM$^2$ Epithelial Voltohmmeter and EndOhm Culture Cup Chamber (WPI, USA). To ensure a fair comparison between measured resistance on-chip and in Transwell, TEER values were normalized to the culture surface areas in both systems and presented as [$\Omega.cm^2$].

To measure epithelium paracellular permeability ($P_{app}$), Cascade Blue tracer dye (Cascade Blue® hydrazide, trilithium salt; Thermo-Fisher, Cat. no. C3239) was introduced to the apical medium at 50 µg/ml and perfused through the epithelial channel of the chip. 40 µl of the apical and basal channels of the chip effluents were collected and their fluorescent intensities (390 nm/420 nm) were measured with a multi-mode plate reader (BioTek NEO, Gen5 3.11) at each timepoint throughout the culture. $P_{app}$ was calculated based on a standard curve and the following equation, as previously described[115]:

$$P_{app} = \frac{V_r \times C_r}{A \times t \times \frac{(C_{d-out} \times V_d + C_r \times V_r)}{(V_d + V_r)}} \tag{1}$$

Where, $V_r$ (mL) is volume of receiving channel at time t (s), $V_d$ (mL) is volume of dosing channel at time $t$, $A$ (cm$^2$) is the area of membrane (0.167 cm$^2$), $C_r$ (µg/mL) is the measured concentration of the tracer in the receiving channel, and $C_{d-out}$ (µg/mL) is measured concentration of tracer in the dosing channel effluent.

## Live mucus imaging on-chip

To visualize live cervical mucus in the Cervix Chip, fluorescent wheat germ agglutinin (WGA) (25 µg/ml, Invitrogen, Cat. no. W11261) was perfused through the apical epithelium channel at 30 µl/hr flow rate for 1 h followed by HBSS wash for 30 min−1 h. The chips were then cut on the sides parallel to the length of the main channel and rotated on one side on a cover glass coated with PBS, additional PBS was applied to the top side of the chip and covered with a cover glass as described previously in[40]. Dark field and fluorescence images were acquired with

an inverted microscope (Axial observer Z1 Zeiss) with LD PlnN 5X/0.4 Ph2 objective lens, and OCRA-Flash4.0 C11440 Hamamatsu digital camera. Quantitative mucus area analysis was performed using Fiji software (V1.8).

## Mucus collection

Cervical mucus was collected using 20mM N-acetylcysteine (NAC) (Sigma, A9165) in DPBS that was perfused though the apical epithelial channel of the chip or added to the Transwell apical chamber and incubated for 3 h before collection. Collected mucus samples were used in total mucin quantification using Alcina blue assay and glycomic profile analysis using Agilent nanoscale liquid chromatography quadrupole time-of-flight coupled to tandem mass spectrometry (nanoLC-QTOF MS/MS).

## Alcian Blue mucin assay

The total mucin content was determined following the method described in[54] by equilibrating the collected chip mucus samples with Alcian Blue stain (ThermoFisher, Cat. no. 88043) for 2 h, followed by centrifuging the resulting precipitant at 1870 g for 30 min, then 2 X wash/spin cycles with a washing solution composed of 40% ethanol, 0.1 mol/L acetic acid, and 25 mmol/L magnesium chloride. The stained mucin pellets were then dissociated in 10% sodium dodecyl sulfate solution (Sigma, Cat. no. 71736), and the absorbance was measured with a microplate reader (Syn-Q55 Synergy HT, BioTek) at 620 nm. Mucin concentrations were obtained based on a curve fitted to mucin standards developed from submaxillary gland mucin (Sigma, Cat. no. M3895) serially diluted from 0 – 500 µg/ml, and Alcian Blue stained as described above.

## Mucus Fern Test

Mucus ferning pattern was qualitatively analyzed by using 10 µl of undiluted apical effluent that was deposited on a micro cover glass (BRAND®, Cat. no. 4708 20). The drop was manually spread using a pipette tip and allowed to dry in room air for 1 h before the ferning patterns were visualized on Revolve microscope (Revolve Software V6.0.1) in the inverted mode.

## Gene expression analysis

Total RNA was isolated from the cell lysate of the cervical epithelial cells of chips and Transwells at day 7 of differentiation, using RNeasy Micro Kit (QIAEGN, Cat. no. 74004), followed by complimentary DNA synthesis using SuperScript VILO MasterMix (Invitrogen, Cat. no. 11755-050). Cellular gene-expression levels were determined using RT-qPCR according to the TaqMan™ fast Advanced Master mix (Thermofisher, Cat. no. 4444963) with 20 µl of reaction mixture containing gene-specific primers for mucin 5B (*MUC5B*, Thermosfisher Scientific, Cat. no. 4331182, Assay ID Hs00861595), mucin 4 (*MUC4*, Thermosfisher Scientific, Cat. no. 4331182, Assay ID Hs00366414), L- asparaginase enzyme (*ASRGL1*, Thermosfisher Scientific, Cat. no. 4331182, Assay ID Hs01091302), and secretory leukocyte peptidase inhibitor (*SLPI*, Thermosfisher Scientific, Cat. no. 4331182, Assay ID Hs01091302). The expression levels of the target gene were normalized to Glyceraldehyde 3-phosphate dehydrogenase (*GAPDH*, Thermosfisher Scientific, Cat. no. 4331182, Assay ID Hs01922876).

To analyze differential gene expression, total RNA samples (150−300 ng/sample) from Cervix Chips and Transwells were submitted to Genewiz commercial sequencing facility (South Plainfield, NJ) for Next Generation Sequencing (NGS). All submitted samples had an RNA integrity number (RIN)> 8.9. Stranded TruSeq cDNA libraries with poly dT enrichment were prepared from total RNA from each sample according to the manufacture's protocol. Libraries for the 37 cDNA samples were sequenced using the Illumina HiSeq sequencing platform yielding 20−30 million 150 bp paired end (PE) sequence reads per sample. Reads have been mapped to Ensembl release 86 using STAR

(v2.5.2b) and read counts have been generated using the feature Counts of Subread package (v2.0.3) and analyzed using a custom bioinformatics workflows implemented in R (v3.6.3). Read counts have been normalized across libraries using the median ratios method implemented in DESeq2 (v.1.26.0), and ggplot2 (v3.3.5) and heatmap (v1.0.12) packages have been used to create figures and heatmaps. GTEx analysis V8 data representing gene expression profiles from human endocervix and ectocervix were obtained from the GTEx Portal (accession no. EGAS00001004439).

## Analysis of cytokines and chemokines
The epithelial effluents from the Cervix Chip were collected and analyzed for a panel of cytokines and chemokine-INF-y, IL-1α, IL-1β, IL-10, IL-6, IL-8, and TNF-α —using a custom U-PLEX® Biomarker Assay kit (Mesoscale Discovery, Cat. no. K15067L-1). The analyte concentrations were determined using a MSD Model 1300 instrument coupled with Discovery Workbench 4.0.12 software.

## Glycomic analysis
*N-glycan release*. The samples were transferred to pre-rinsed Amicon Ultra-0.5 (10 kDa) centrifugal filter units (MilliporeSigma, MA) and cleaned up with 400 μL of nanopure water followed by centrifugation at 14,000 × g for 15 min three times to remove salts. The protein samples were then recovered by reverse-spinning the unit at 1000 × g for 2 min. As described previously[116], dithiothreitol (DTT) and $NH_4HCO_3$ were added to the purified samples to final concentrations of 5 mM and 100 mM, respectively. The mixture was heated in a boiling water bath for 2 min to denature the proteins. *N-glycan release*. 2 μL of PNGase F (New England Biolabs, MA) was added, and the samples were incubated at 37 °C in a microwave reactor (CEM Corporation, NC) for 10 min at 20 W. The samples were further incubated in a 37 °C water bath overnight to fully convert the glycan's amine group to hydroxyl group. After the incubation, 350 μL of nanopure water was added and ultracentrifuged at 200,000 × g for 45 min at 4 °C. The protein pellets were saved for O-glycan release. The supernatant containing the released N-glycans was desalted with porous graphitized carbon (PGC) solid-phase extraction (SPE) plates (Glygen, MD). The PGC-SPE plate was activated by 80% (v/v) acetonitrile (ACN) and equilibrated with water prior to use. The N-glycans were washed with 3 vol. of water, eluted with 40% (v/v) ACN, and dried in vacuo. *O-glycan release*. The de-N-glycosylated protein pellets were resuspended in 90 μL of nanopure water by sonication for 20 min. 10 μL of 2 M NaOH and 100 μL of 2 M $NaBH_4$ were added and the mixture was incubated in a 45 °C water bath for 18 h. The reaction was quenched by mixing with 100–120 μL of 10 % acetic acid on ice until the pH values reached 4–6. Then the samples were centrifuged at 21,000 × g for 30 min at 4 °C. The supernatant containing the released O-glycans was desalted with PGC-SPE plates (Glygen, MD), and further purified with iSPE-HILIC cartridges (Nest Group, MA). The HILIC cartridges were activated with ACN and water and equilibrated with 90% (v/v) ACN. The dried O-glycan eluates from PGC SPE were reconstituted in 90% (v/v) ACN and allowed to pass through the HILIC cartridges five times. The O-glycans were then washed with 5 vol. of 90% (v/v) ACN, eluted with water, and dried in vacuo.

Glycomic analysis with LC-MS/MS was employed as previously described[61]. In general, the glycan samples were reconstituted in 8 μL of nanopure water and analyzed with an Agilent 1200 series HPLC-Chip (PGC) system coupled to an Agilent 6520 Accurate-Mass Q-TOF MS (Agilent, CA). The glycans were first loaded onto the chip with aqueous mobile phase A, 3% (v/v) ACN and 0.1% (v/v) formic acid (FA) in water, at a flow rate of 3 μL/min, and then separated using a binary gradient with the same mobile phase A and organic mobile phase B, 90% (v/v) ACN and 1% (v/v) FA in water, at a flow rate of 0.3 μL/min. The binary gradient was 0.0–2.5 min, 1% (B); 2.5–20.0 min, 1–16% (B);

20.0–35.0 min, 16–58% (B); 35.0–40.0 min, 58–100% (B); 40.0–50.0 min, 100–100% (B); 50.0–50.1 min, 100–1% (B); 50.1–65.0 min, 1–1% (B). The drying gas temperature and flow rate were set at 325 °C and 5 L/min, respectively. The capillary voltage was adjusted between 1850–2000 V to maintain a stable electrospray. MS spectra were acquired over a mass range of *m/z* 600–2000 for N-glycans and *m/z* 400–2000 for O-glycans in positive ionization mode at a scan rate of 1.25 s/spectrum. Four most abundant precursor ions were fragmented through collision-induced dissociation (CID) with nitrogen gas. MS/MS spectra were collected over a mass range of *m/z* 100–2000 at 1 s/spectrum.

Glycan compound chromatograms were extracted from the raw data using the MassHunter Qualitative Analysis B.08 software (Agilent, CA) with in-house libraries containing common N- and O-glycan accurate masses. N-glycans were identified using the Find by Molecular Feature algorithm with a 20 ppm mass tolerance, and O-glycans the Find by Formula algorithm with a 10 ppm mass tolerance. The glycan structures were confirmed by tandem MS spectra. The relative abundance of each glycan was calculated by normalizing its peak area to the total peak area of all glycans identified.

## Proteomic analysis
Digested epithelial cells from the chip were dissolved in 5% SDS (Thermo, CA) and loaded on a micro S-trap column (Protifi, NY) and digested according to vendor protocol for 1 h at 50 C by Trypsin Platinum (Promega, WV). Digested material was then eluted from the column by centrifuge at 123 x g rpm for 2 min and dried in speedvac (Eppendorf, Germany). Each prepared sample was separated on Hi-pH column (Thermo, CA) according to vendor instructions. After separation each fraction was submitted for single LC-MS/MS experiment that was performed on a HFX Orbitrap (ThermoScientific, Germany) equipped with dual 3000 nano-HPLC pump (ThermoScientific, Germany). Peptides were separated onto a micropac 5 cm trapping column (Thermo, Belgium) followed by 50 cm micropac analytical column 50 cm of (ThermoScientific, Belgium). Separation was achieved through applying a gradient from 5–27% ACN in 0.1% formic acid over 180 min at 200 nl min−1. Electrospray ionization was enabled through applying a voltage of 1.8 kV using a home-made electrode junction at the end of the microcapillary column and sprayed from stainless-steel 4 cm needle (ThermoScientific, Denmark). The HFX Orbitrap was operated in data-dependent mode for the mass spectrometry methods. The mass spectrometry survey scan was performed in the Orbitrap in the range of 450 –1,800 m/z at a resolution of $1.2 \times 10^5$, followed by the selection of the ten most intense ions (TOP10) for HCD-MS2 fragmentation in the orbitrap. The fragment ion isolation width was set to 0.7 m/z, AGC was set to 50,000, the maximum ion time was 200 ms, normalized collision energy was set to 27 V and an activation time of 1 ms for each HCD MS2 scan.

Raw data were analyzed using Proteome Discoverer 2.5 (Thermo Scientific, CA) software. Assignment of MS/MS spectra was performed using the Sequest HT algorithm by searching the data against a protein sequence database including all entries from the Human and bacteria Uniprot database[117] as well as known contaminants such as human keratins and common lab contaminants. Sequest HT searches were performed using a 10 ppm precursor ion tolerance and requiring each peptides N-/C termini to adhere with Trypsin protease specificity, while allowing up to two missed cleavages. 18-plex TMTPRO tags on peptide N termini and lysine residues ( + 304.207146 Da) was set as static modifications while methionine oxidation ( + 15.99492 Da) and asparagine and glutamine deamidations ( + 0.984016 Da) were set as variable modification. A MS2 spectra assignment false discovery rate (FDR) of 1 % on protein level was achieved by applying the target-decoy database search. Filtering was performed using a Percolator (64 bit version[118]). For quantification, a 0.02 m/z window centered on the

theoretical m/z value of each the eighteen reporter ions and the intensity of the signal closest to the theoretical m/z value was recorded. Reporter ion intensities were exported in result file of Proteome Discoverer 2.5 search engine as an excel tables. The total signal intensity across all peptides quantified was summed for each TMTPRO channel, and all intensity values were adjusted to account for potentially uneven TMT labeling and/or sample handling variance for each labeled channel. Protein level data from each TMT channel was analyzed in R. The dataset was first preprocessed by removing contaminant proteins and those with unavailable abundance data. Next, the dataset was normalized using the variance stabilizing normalization method (vsn) with the R MSnbase package.

## Bacterial co-culture on-chip

To mimic and study the co-culture of an optimal vaginal microbiota on the Cervix Chip, a consortium of three *L. crispatus* strains (C0059E1, C0124A1, C0175A1) was constructed that were originally cultivated from women with stable *L. crispatus* dominant microbiota participated in UMB-HMP study[72]. To mimic and study the infection of a non-optimal vaginal microbiota, a *Gardnerella vaginalis* (*G. vaginalis*) consortium was constructed from two *G. vaginalis* strains (*G. vaginalis* E2 and *G. vaginalis* E4), originally isolated from women with diverse, non-*L. crispatus* consortia participated in UMB-HMP study[72]. After the Cervix Chip was matured and functionally stable at day 12 of culture, the chip was inoculated with the bacterial communities and co-cultured for additional 3 days. Infection inoculum of *L. crispatus* and *G. vaginalis* consortia were made by combining required volumes of each strain to create equal cell density of each *L. crispatus* or *G. vaginalis* strain per 1 ml of inoculum. The bacteria were washed and resuspended in HBSS (LB/ + G) to make 2 X concentration inoculum on ice. The Cervix Chip mucus was collected from the chips and mixed with the 2X inoculum (1:1) to make the 1X inoculum. Apical epithelium channel of the Cervix Chip was inoculated with *L. crispatus* consortia at $5.9 \times 10^6$ CFU/ml or with *G. vaginalis* consortia at $1.9 \times 10^6$ CFU/ml. The infected chips were maintained under static culture for 20 h at 37 °C and 5% $CO_2$ followed by 72 h of co-culture under episodic apical flow and continuous basal flow on the Zoe culture module. At 24-, 48-, and 72 h post inoculation, the Cervix Chip was perfused apically with customized HBSS (LB/ + G) for 4 h at 30 µl/hr to quantify the non-adherent bacterial CFU in the chip effluents at each timepoint. The adherent bacteria were measured at 72 h post inoculation within the chip epithelial cells digested with Collagenase IV (Gibco, Cat. no. 17104019) in TrypLE (Thermo Fisher, Cat. no. 12605010) at final concentration of 1 mg/ml for 1.5 h at 37 °C and 5 % $CO_2$.

Bacterial strain infection stocks were made following the method described in Mahajan et al.[42]. Colony forming unites (CFU)/ml of the bacteria collected from the chip effluents and epithelial cell digest was enumerated by spread plating the samples on MRS agar (Hardy, Cat. no. G117) for *L. crispatus* samples and on Brucella blood agar (with hemin and vitamin $K_1$) (Hardy, Cat. no. W23) for *G. vaginalis* samples under anaerobic conditions. The colonies were counted after 72 h of incubation at 37 °C and CFU/mL was calculated for each species.

The number of epithelial cells in the Cervix Chip was obtained by digesting the Cervix Chip apical channel at day 7 of differentiation or day 3 on co-culture with bacteria with Collagenase IV (Gibco, Cat. no. 17104019) in TrypLE (Thermo Fisher, Cat. no. 12605010) at a final concentration of 1 mg/ml for 1.5 h at 37 °C and 5% $CO_2$. The total number of cells was quantified in the collected cell suspension using Trypan Blue staining assay and Hemocytometer.

## Statistics and reproducibility

All results are presented from at least two independent experiments and data points are shown as mean ± standard error of the mean (s.e.m.) from multiple experimental replicates or Organ Chips. Statistical analysis was performed using GraphPad Prism version 8.1.2 (GraphPad Software Inc., San Diego, CA) and the statistical significances between groups were tested using two-tailed Student's *t*-test or One-way ANOVA with Bonferroni correction for multiple hypothesis testing, and all *P*-values are presented in the figures. Quantitative image analysis was done using Fiji version (V1.8).

## Reporting summary

Further information on research design is available in the Nature Portfolio Reporting Summary linked to this article.

## Data availability

The sequencing data reported in this paper have been deposited in the GEO database under accession no. GSE231016. Gene expression profiles from human endocervix and ectocervix were obtained from GTEx analysis V8 data accessed on GTEx Portal (https://www.gtexportal.org, accession no. GSE168244) on 01/05/2022. The raw glycomic data reported in this paper can be found on the MassIVE repository [https://doi.org/10.25345/C52805830, accession no. MSV000091806]. The raw proteomic data generated in this study are deposited at MassIVE repository [https://doi.org/10.25345/C5F18SR7P, accession no. MSV000093636]. The used human and bacteria protein sequence databases are accessible at Uniprot database [human in UniProtKB search (51829) | UniProt and bacteria in UniProtKB search (341308) | UniProt]. All other data reported in this study are available within the paper, its Supplementary Information or Source Data files provided with this paper. Source data are provided with this paper.

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

## Acknowledgements

We thank T. Ferrante for his guidance with microscopy imaging and analysis, D. Chou for providing clinical mucus samples, A. Naziripour, B. Lobamba, and R. Prantil-Baun for their help and inputs in the early stages of the project, and Drs. J. Ravel and S. Rakoff-Nahoum Lab for providing the initial strains of *L. crispatus* and *G. vaginalis*. We acknowledge research funding from Bill and Melinda Gates Foundation (INV-035977 to D.E.I.), Canada's Natural Science and Engineering Research Council (NSERC) (to Z.I.), and the Wyss Institute for Biologically Inspired Engineering at Harvard University (to D.E.I.).

## Author contributions

Z.I. and D.E.I. conceived this study. Z.I. designed, performed and analyzed experiments with other authors assisting with experiments and data analysis. J.C. assisted with chip experiments and microscopy imaging and analysis. C.B.L. developed the glycomics mass spectrometry method, and S.C. and Y. X. performed the associated experiments and analysis. V.H. performed RNA-seq analysis. A.S. performed mucus Fern Test and Second Harmonic imaging. A.G., N.T.L. and E.R.D. assisted in bacterial studies. B.B. and S.S. performed proteomic mass spectrometry experiments and analysis. T.T. generated Pseudo-H&E images. S.E.G. assisted with RNA extraction and RT-qPCR. A.M.S. assisted with sensor-integrated chip studies. S.E.G., A.M.S. and G.G. managed the project progress. Z.I. and D.E.I. wrote the manuscript with all authors providing feedback.

## Competing interests

D.E.I. holds equity in Emulate, chairs its scientific advisory board and is a member of its board of directors. The remaining authors declare no competing interests.
