## [Peer Review File · Nature Communications]

REVIEWER COMMENTS

Reviewer #1 (Remarks to the Author):

NCOMMS-23-17604

Mucus production, host-microbiome interactions, hormone sensitivity, and innate immune responses modeled in human endo- and ectocervix chips.

The manuscript by Izadifar et al. describes the development of the cervix on a microfluidic chip that shows superior characteristics and performance compared to a transwell system that can hold two cell types. Ecto/endo cells cultured together, and stromal cell interface produce mucin, demonstrating immune response and sensitivity to hormones. Overall, the manuscript is well written and contains a lot of advanced technologies besides the OOC model to evaluate various physiologic and pathologic parameters indicative of an unhealthy cervix. Several concerns persist, and I like to have authors address some of my concerns before consideration for publication. The chip used for this project is a commercially available two-chamber chip used for modeling different organs and disease processes. The primary purpose of this study is to demonstrate the superiority of a chip-based approach over a transwell system restricted to specific environmental conditions and limited manipulability. My primary concern is that the combination of ecto and endo cervical cells and the addition of stromal cells in a separate chamber does not necessarily reflect the architecture of the cervix, and authors tend to undermine the structural complexity of the cervix in evaluating its property. This is an overly simplified version of the cervix tissue, maybe slightly superior to a transwell system. This is a significant limitation for gynecologists and obstetricians, and this device also fails to address a clinician's existing fears of cell type-specific contribution to the pathology of the organ.

My other major concerns are listed below:

1. Authors have used commercially available ecto and endo cervical cells and cultured them together. This is a limitation as the behavior of these cells and their response to immune and endocrine mediators (that were determined as outcome measures in this manuscript).
 - a. Fig 1 shows the development and characteristics of the chip (and cellular transcriptome) and its comparison with transwell data. Please indicate the anatomical structure replicated in the transwell and on the chip.
 - b. Have these data been compared to the cervical tissue transcriptome to show the similarity of this model to the actual cervix?
 - c. What is the rationale for a continuous flow? The cervix is not experiencing a dynamic flow situation, as depicted in the approach section of this manuscript.

2. Anatomically ectocervix is microbe-laden and the endocervix constitutes the sterile component. These are separated by transformation zone that is constantly changing cells.

3. The ectocervix is highly resistant to exogenous factors in cervical biology, whereas the endocervix is an extremely vulnerable component. The authors have compromised the true functional contributions of each cell type by combining the two layers. The parameters (mucus production, host-microbiome interactions, hormone sensitivity, and innate immune responses) modeled are not the same when a transformation zone separates these layers, and the impact of stroma on each cell type is distinct.

a. Please refer to reviews and contributions by Mahendroo et al. I to the field. Unfortunately, the authors fail to reference a single paper by Dr. Mahendroo

4. Authors use the term 'epithelial cell differentiation.' What exactly is referred to here? Are these cells expected to transition into stromal cells, or are they expected to differentiate into a different cell type? What are the pertinent markers indicative of differentiation used to identify them? These data should be included.

5. Please elaborate on the conditions and how the authors prevented ecto and endo cervical cell transition into stromal phenotype in their culture setups.

6. The emphasis is on the biophysical properties of mucus (Fig 2). This is absolutely one of the major contributors to cervical physiology and function. Experiments used to determine the physiochemical properties are lauded. A lot of experiments have been dedicated to demonstrating the importance of mucin, its detectability on a chip, their changes and chemical composition under different environment are extremely valuable. One wonder why the authors used flow conditions to show mucus production. MUC5A has been produced by cervical cells in a 2D culture setup even without flow.

7. One of the major constituents of the cervical stroma are immune cells. Manuscript does not mention the contributions of immune cells. It is unclear how innate immune response is exclusively becomes the property of epithelial and stromal cells. This is a limitation of this model.

8. The contribution of immune cells to biology is well reported in the cervical functions PMID: 24410939 PMID: 30629144 PMID: 28395330 PMID: 25811906 PMID: 19234164 PMID: 26637953.

9. Stroma with macrophages has a well-defined role on microbial response and their proliferations and transitions are major contributors of cervical mucin biology, matrix remodeling and dysfunctions during dysbiosis.

10. Please indicate the endogenous production and contribution of estradiol-17beta in your model or other endocrine factors. What is the need for exogenous supply?

11. Page 7; lines 144-145 – Why there is a decline in cell proliferation? How cellular stress of flow and or long term culture impact the cells?

12. Page 8; line 168 – is it possible that cytokines are washed out due to continuous flow? This is still not a physiologically valid model.

13. Page 10; If the ecto-cervical region is heavy in N-glycans does that mean it is not clinically and or physiologically relevant? Is this a limitation of the model? How does that translationally impact cervical function?

14. The manuscript is largely and primarily focused on mucus production but relates poorly to cellular changes. Please provide emphasis to cellular changes as the multiplicity of cells makes cervix a unique organ and not just mucus.

15. Page 14; line 322– such a flow of microbiome to impact endocervix does not exist in vivo. Endo is not in direct contact to the VEC or the microbiome. Ecto is and hence, the role of ecto to microbes and their repercussion on endo is more critical. See my comments above.

16. Page 14; line 323 – "this would have the highest relevance for the health of the upper reproductive tract". This is not true. See my comments above. Please rewrite this sentence.

17. Page 14; line 333 - It seems that 3 days is the limit for these cultures. The rest of the experiments are all 12 days. Please clarify as I am bit confused with the number of days in culture environment and for various experiments.

18. Fig 4; please explain if the vaginal microbiome interaction with vaginal epithelial cell has been factored in the discussion. These microbes, if any, interact with ecto cells before impacting endo cells. The metabolites of the microbiome – vaginal cell interactions produce most of the repercussion effect in subsequent upper genital tract layers.

19. Please include data related to collagen production by stromal and epithelial cells. ECM coating was mentioned (page 5) but the changes to this component was not mentioned in response to experimental conditions.

Minor concerns:

Authors should not try stratified analysis based on race with limited samples available. Without replication and validation, these data are merely misleading. The discussion on this topic should be avoided.

The comparison of cervical cell based data to airway epithelial cell is not a valid comparison. Please remove those references throughout the manuscript.

Key references of people who contributed to the field of cervical biology (Mahendroo et al, Ward et al, Stock et al, Yellon et al) are missing. Their work should be recognized.

Reviewer #2 (Remarks to the Author):

In this study by Izadifar and colleagues, the Emulate microfluidic platform was used to create human endo- and ectocervix tissues-on-a-chip. They established the model using two different epithelial cell lines – one from endocervix and the other from ectocervix. The underlying support cells, the stromal cells were also included. Two different flow patterns, continuous and periodic were studied and multiple endpoints of cervix function was assessed, in the presence of steroid hormones as well as bacteria. This

is a well written manuscript with state of the art technologies and presents a new in vitro model for the human cervix. This model can be used to study BV, and test drugs. The novelty and physiologic similarity of the system to in vivo tissues, makes this study one of high impact. However, there are clarifications needed, use of incorrect terminology as well as lack of power for race specific data. The concerns are indicated below.

Major concerns:

- The rationale for mixing endo- and ecto-cervical epithelial cells is not provided. The endocervix and ectocervix are very different with different physiology. It was unclear why both were mixed together. Furthermore, the importance of the presence of stromal cells was not studied. Why were these cells included and what changes were observed in each of the experiments that were done? Given that continuous versus periodic flow rates promoted enrichment of different epithelial cells, all differences in functional measurements could simply be due to the different population of cells and not necessarily the impact of flow on the cells. Control experiments using just one cell epithelial cell type would promote a greater understanding of the impact of flow and shear stress.
- Why were two different flow rates studied? Is this physiologically relevant? There is no continuous flow of liquid that the endo- and ecto-cervix is exposed to in vivo.
- Rather, mechanical stimulation to mimic uterine contractions would have been more physiologic.
- *note that “ovulatory” vs “non-ovulatory” should not be used in this study to describe E2 vs E2+P4 treatment. Please use “follicular” for E2 treatment and “luteal” for E2+P4 treatment. Non-ovulatory is misleading and the field does not describe the hormonal stages of the cycle as such.
- Throughout the manuscript the term “differentiation” is used. This implies that there are progenitor cells that differentiate to endo or ectocervix epithelium, and it is assumed that this is not the case of the cell lines that you are using. You are treating the cells and the cells are responding to the stimulus. Cells are not differentiating. Please correct.
- The data showing race differences, Black, White and Hispanic is meaningless as N=1 for each race. Without sufficient power, it is not known whether race is the determining factor for different gene expression or whether it is reflecting the heterogeneity that naturally occurs between patients even within the same race. Furthermore, self reported race is not entirely accurate and at least for the Black population, ancestry analysis would confirm race. In addition, the race of the stromal cells was not determined.
- As PDMS is known to absorb hydrophobic molecules, it would be important to measure E2 and P4 concentrations in the media after flowing through the device.

Specific comments:

Line 24 – the authors state that they recreate the cervical epithelial-stromal but have no data to show the role of the stroma. How does the interface work mechanistically? After Fig 1F there is little to no reference to the contributions of the stromal cells to the correct functionality of the epithelial cells. Is there anything known about their use or function? It would be important to see the mucus production in

periodic and continuous flow without the stromal cells present. Otherwise, this model could have been simplified.

Line 114 – Please explain which flowrates were used , ie, why only 40ul/h is optimal and why intermittent used 25% less flowrate for 4 hours.

Line 118 – Other systems did not continuously perfuse the stromal layer because these cells usually do not experience shear stresses – how is this 40ul/hr in a 1mm by 0.2mm channel physiological? What would happen with less flow in stromal cells?

Fig 1A – Add the term apical and basal to image.

Fig 1D – Indicate the confines of the channel.

Line 132-138 and Fig 1E-D: The authors have done a good job demonstrating that the cervical epithelial cell layer presents cervical epithelium-specific markers, however no comment on the morphology of the construct is made. The concern here is that they used a commercial mix of epithelial cells from the endo- and ectocervix, which have very different morphologies in vivo (stratified squamous and columnar mucinous epithelium respectively). From the images it looks like the epithelial construct is only a couple of cell layers thick with no tissue-specific morphology. Please elaborate in the manuscript.

Line 148 & Fig s1b – is this different expression observed in both flow conditions? Are the intermittent and continuous results averaged?

Line 158 & Fig s1e – Barrier function was measured using TEER and chip vs Transwell resistance was measured. Can you elaborate on what kind of Transwell was used, and if it also had an adherent layer of epithelial cells and stromal cells divided by a permeable membrane? This information is also useful to add to the methods in line 479. Lastly, assuming that the cell numbers and surface areas are very different between transwells and chips, is there a type of normalization necessary to be able to compare these cultures?

Line 168 – is the decrease in IL-8 and TNF-a is due to less mechanical stimulation or due to the accumulation of waste products during the 20 hour static period.

Line 170 – Given the different cell populations, it would have been more informational to have done single-cell RNA sequencing and demonstrate/cluster the survival of specific cell populations in periodic or continuous flow. Is there any way to isolate relative quantities of endo and ectocervical epithelial cells from this data?

Line 221 – this is the first time that endo- and ecto-chip has been used, and should be defined here.

Line 237 – Is the clinical mucus considered as a mixture of endo and ecto-cervix excretions?

Line 241 – Please add that permeability plays a critical role during reproduction and allowing the sperm through.

Line 302 – Does a decrease in these N-glycans have an effect on the mucus permeability, has that been described?

Line 325 – Please describe why estrogenic conditions were used.

Reviewer #3 (Remarks to the Author):

In this study, the authors developed an organ-on-a-chip model from human cervical cells that can resemble the physiological features of the cervix including mucin production and also the expected outcomes of interactions with commensal and pathobiont bacteria. The study provides a new tool that can be used to study the health and pathophysiology of the female genital tract in vitro, therefore its use can lead to identification of causes of cervical dysbiosis and development of new therapies.

Here are some minor comments that needs to be addressed:

-The title implies there were two kinds of chips develop: one with endo-cervical cells and one with ecto-cervical cells and this is misleading. Better to simplify it as cervix chips. Alternatively, periodic and continuous flow can be integrated to the title.

- How were exo and endo cervical cells mixed? Same ratio? Why were they mixed instead of colonizing different parts of the chip? What was the reasoning for not studying them separately? Please justify the decision. Why only cervical stromal fibroblasts isolated from the donor and not the cells to match the donors?

-Lines 97-98. The statement to describe endo and ecto cervix for non-experts can be introduced earlier in the introduction. Where those cells are located, their function, and relative importance.

-Line 79 and Lines 373-374. The 3D models such as RWV can capitate the barrier properties and mucus production as well, so this statement needs to be corrected.

-Line 158: please also report the electrical resistance of transwell cells to see the magnitude for the comparison

-Line 170: hyphen between RNA and seq

-Line 180: briefly described the differences observed on endocervical features

-There is a limitation based on lack of biological replicates of donor samples (based on ethnicity) so any comparison to the in vivo is premature. The trends observed can be only defined as a case study. Please tone it down the implications.

Figure 1h. the bar graphs do not align. How many chips were used for each condition? Is it three? Please include on the figure legend.

Figure 2b: the top of the boxes for O-glycans and N-glycans do not align

Line 1002: What does biological chip replicate mean? Does it mean different donor? or are they experimental replicates?

Figure 3. The microscopy images lack the scale information.

Line 714. The sentence ends abruptly, maybe also mention the author name at the end.

RESPONSE TO REVIEWERS

REVIEWER #1:

1. My primary concern is that the combination of ecto and endo cervical cells and the addition of stromal cells in a separate chamber does not necessarily reflect the architecture of the cervix, and authors tend to undermine the structural complexity of the cervix in evaluating its property. This is an overly simplified version of the cervix tissue, maybe slightly superior to a transwell system. This is a significant limitation for gynecologists and obstetricians, and this device also fails to address a clinician's existing fears of cell type-specific contribution to the pathology of the organ.

The goal of this work was not to engineer an artificial cervix, but rather as stated in the title of our article, to develop a more clinically relevant *in vitro* model of mucus production, host-microbiome interactions, hormone sensitivity, and innate immune responses in human cervical tissues than is currently available today. Our data clearly show that these chips do indeed provide a useful model that enables production of mucus with physical and chemical properties more like that observed *in vivo* and we can co-culture microbiome in direct contact with human cervical epithelium over days *in vitro*. This latter capability - the ability to study host-microbiome interactions within a human relevant cervical microenvironment over extended times - is not possible with existing static culture systems, such as Transwells. This is not a minor advance; this enables an entirely new path of investigations, which can lead to new insights into cervical health and disease as well as provide a human preclinical model for evaluation of potential antibiotics or live biotherapeutic products. In fact, to suggest that this would not be useful for gynecologists and obstetricians is not correct as we are currently using these chips in Gates Foundation-funded projects with clinicians around the world to validate live biotherapeutic products *in vitro* before they move into a human clinical trial for bacterial vaginosis. The interest among these clinicians and desire to gain access to this technology is huge because these types of studies studying human cervical response to living microbes over days in culture is not possible using any other *in vitro* model.

We also did not claim to have replicated the full structural complexity of particular regions of the cervix nor did we set out to do this, but our data do demonstrate that we have created a more physiologically relevant *in vitro* model for study of these particular cervical responses. Finally, it is important to note that we carried out these studies using a commercial supply of human cervical epithelial cells, which we discovered contain a mixture of endo- and ecto-cervical cells; we did not intentionally mix these cell populations. Moreover, one of the other novel findings of our study (i.e., in addition to modeling mucus biology and the other responses delineated in our title) that could not be observed in conventional static cultures systems is that different flow conditions push these populations preferentially along either ecto- or endo-cervical differentiation pathways as indicated by our transcriptomic profiling studies. So there are indeed unique insights that can be made using this *in vitro* model even though it is 'over simplified'. In any case, we now explain both the advantages and limitations of our model more clearly in the revised Discussion.

2. Authors have used commercially available ecto and endo cervical cells and cultured them together. This is a limitation as the behavior of these cells and their response to

immune and endocrine mediators (that were determined as outcome measures in this manuscript).

A key goal of this study was to demonstrate the advantages it provides by benchmarking it against existing common *in vitro* organotypic cervical culture models. While most past cervical cultures used established cervical epithelial cell lines, more advanced versions used a commercial source of primary human cervical epithelial cells cultured in Transwell inserts (Zhang et al., *Oncol Lett.*, 2020; Caven et al., *Front Cell Infect Microbiol*, 2023). Thus, we chose to use the same commercial source of healthy primary human cervical epithelial cells that were used in these previous studies (which are only sold by two suppliers) for our studies. The vendor did not provide any technical information about the specific zone of the cervix from which the cells were isolated. In the course of characterizing these cells, we discovered that they contain a mixed population of both endo- and ecto-cervical cell types (presented in **Suppl. Fig. S1a**). We now explain this more clearly in the revised manuscript. However, we also discovered that this caveat had unexpected value in that we could push these cells preferentially down ecto- or endo-cervical differentiation paths using different flow conditions. This is, in itself, a novel finding of this work that could not be uncovered using conventional static systems. But more importantly, this project started by exploring whether the microfluidic Organ Chip model could provide a more physiologically relevant experimental system to study cervical mucus that plays a key role in host-microbiome interactions, and as described above in response to question 1; our results clearly show that it does.

3. Fig 1 shows the development and characteristics of the chip (and cellular transcriptome) and its comparison with transwell data. Please indicate the anatomical structure replicated in the transwell and on the chip.

We are not trying to replicate the full 'anatomic structure' of any particular part of the cervix as we lack endothelium, nerve cells, lymphatics, etc. We are simply recreating a cervical epithelial-stromal interface in both the chip and Transwell to enhance epithelial differentiation, and we add fluid flow in the Cervix Chip. The phase and immunofluorescent microscopic images of the cervical epithelium and interfaced stromal layer formed on the chip as compared to the Transwell inserts are shown in **Supp. Fig. S1g**.

4. Have these data been compared to the cervical tissue transcriptome to show the similarity of this model to the actual cervix?

The genes shown in the RNA-seq analysis in **Fig. 1i** were selected based on a recent publication (Lohmussaa et al., *Cell Stem Cell*, 2021) that identified and reported this set of genes as differentially and distinctly regulated in the transcriptome of human ecto- versus endo-cervical tissues *in vivo*. Therefore, we used the same gene signatures for analysis of the Cervix Chip transcriptome and showed its similarity to that of the human tissues from the different cervix regions. This also revealed the effect of the continuous versus periodic fluid flow on differentiation along ecto- versus endo-cervical paths on-chip.

5. What is the rationale for a continuous flow? The cervix is not experiencing a dynamic flow situation, as depicted in the approach section of this manuscript.

Although human cervix is not perceived to experience a significant form of fluid flow similar to that in the intestine or arteries, mucus is continually produced, hydrated by interstitial

fluids, and cleared from the surface of the epithelium as it moves (flows) to the vagina by mucociliary transport. This can be further augmented by gravity and by changes in endocrine hormones and other microenvironmental factors that influence mucus production in the reproductive tract. This flow, even if minimal, can influence epithelial cell function through associated application of fluid shear forces or other physical forces (e.g., osmotic) just as it does in many other organs. In a culture system such as this, the continuous removal of mucus, preventing it from accumulating abnormally, may be equally important to mimic the *in vivo* condition. As such, we believed that it is physiologically relevant that the cervical epithelium surface experience a gentle continuous or a less frequent periodic flow associated with propelling mucosal discharge. Because this was the first effort for developing the Cervix Chip and little is known about physiological mucus flow conditions *in vivo*, we did not know which flow condition would be optimal for promoting the growth and maturation of cervical epithelial cells. So, we tested both continuous and periodic flow conditions that mimic frequent and infrequent mucosal discharge flow, respectively, *in vivo*. This effort led to the interesting and novel observation that different flow conditions moved the mixed cervical epithelial population along different differentiation paths (ecto- vs. endo-cervical), which we report here. We now clarify our rationale and explain these points more clearly in the manuscript.

6. Anatomically ectocervix is microbe-laden and the endocervix constitutes the sterile component. These are separated by transformation zone that is constantly changing cells.

Although the endocervix and other organs of the upper reproductive tract had been long presumed to be sterile microenvironments, a recent study by Chen et al. 2017 published in this journal (*Nature Comm.*; DOI: [10.1038/s41467-017-00901-0](https://doi.org/10.1038/s41467-017-00901-0)) systematically studied the microbiota composition of the female reproductive tract in 110 reproductive age women. This study revealed the presence of commensal bacterial communities in the cervical canal, uterus, and fallopian tubes as well as in the vagina. Thus, the endocervix is not a 'sterile component'. Although there is a transformation zone between the ecto- and endo-cervix, histology shows that there is a discrete transition between these two regions (i.e., between stratified and columnar epithelium) *in vivo*.

7. The ectocervix is highly resistant to exogenous factors in cervical biology, whereas the endocervix is an extremely vulnerable component. The authors have compromised the true functional contributions of each cell type by combining the two layers. The parameters (mucus production, host-microbiome interactions, hormone sensitivity, and innate immune responses) modeled are not the same when a transformation zone separates these layers, and the impact of stroma on each cell type is distinct.

We appreciate your point and acknowledge the complexity of the cervix in terms of localized tissue phenotypes with different biological responses. However, the focus of this work was not to develop zone-specific cervical tissues, nor did we 'combine the two types'. As described earlier, in the course of these studies we discovered that commercial cell source of the 'cervical epithelial cells' that they sold contained a mixture of ecto- and endo-cervical cells. Although we describe this as a potential limitation in our Discussion, we also were able to take advantage of this complexity to learn that different flow environments can push these cells down

distinct ecto- vs. endo-cervical lineages. Moreover, by using these different flow conditions, we were able to generate Cervix Chips that exhibited either ecto- or endo-cervical phenotypes, and demonstrated that these more closely mimic *in vivo* cervical mucus physiology than other *in vitro* Transwell models currently used in this field. We showed that these two cervical phenotypes on-chip exhibit different mucus composition (mucin glycosylation) and innate immune responses, supporting the point that this Reviewer raises here.

8. Please refer to reviews and contributions by Mahendroo et al. I to the field. Unfortunately, the authors fail to reference a single paper by Dr. Mahendroo

Thank you for pointing this out. We have now included contributions from Dr. Mahendroo's work in the manuscript.

9. Authors use the term 'epithelial cell differentiation.' What exactly is referred to here? Are these cells expected to transition into stromal cells, or are they expected to differentiate into a different cell type? What are the pertinent markers indicative of differentiation used to identify them? These data should be included.

Primary cervical epithelial cells predominantly proliferate in conventional 2D cultures and express cytokeratin 5 (ck5) and Ki67 at high levels, but low levels of mucin type 5B (MUC5B), the major gel forming mucin in cervical mucus *in vivo*. In contrast, the human cervix *in vivo* is mostly constituted of non-proliferating, mucus-producing (MUC5B+) epithelial cells that only have proliferating ki67 positive cells sparsely distributed in the basal layer of the tissue. We seeded the chips with the cervical epithelial cells that had been expanded in the 2D culture and then used an optimized medium and culture condition to induce differentiation (maturation into a tissue composed of fewer proliferating cells, which is dominated by mucus producing cells) on-chip over 12 days of culture. We quantified Ki67 and MUC5B expression on-chip to determine the abundance of the cells representing these markers in the epithelium throughout the culture. The data presented in the **Supp Fig. S1d** shows that the abundance of proliferating cells (Ki67⁺) and mucus producing cells (MUC5B⁺) significantly reduces and increases, respectively, demonstrating increased differentiation over time in culture indicating formation of mature cervical epithelium in the Cervix Chip that more closely resembles the cervical epithelium *in vivo*.

10. Please elaborate on the conditions and how the authors prevented ecto and endo cervical cell transition into stromal phenotype in their culture setups.

Epithelial-mesenchymal transition (EMT) is a transdifferentiation process in which epithelial cells adopt characteristics commonly found in mesenchymal cells. In our model, the epithelial cells are constantly exposed to sex hormones to prevent EMT. Our lab and many others have also shown that culturing living human tissues under dynamic fluid flow in Organ Chips enhances tissue-specific differentiation in many organ models (Ingber, Nature Rev. Genetics 2022, DOI: [10.1038/s41576-022-00466-9](https://doi.org/10.1038/s41576-022-00466-9)). This is consistent with our finding that culturing the cervical epithelium promotes expression of tissue-specific differentiation functions, including mucus (MUC5B) production as well as expression of estrogen and progesterone receptors and cytokeratin 18 that are not typically associated with mesenchymal cells or fibrosis processes. We now explain this more clearly in the Results.

11. The emphasis is on the biophysical properties of mucus (Fig 2). This is absolutely one of the major contributors to cervical physiology and function. Experiments used to determine the physiochemical properties are lauded. A lot of experiments have been dedicated to demonstrating the importance of mucin, its detectability on a chip, their changes and chemical composition under different environment are extremely valuable. One wonder why the authors used flow conditions to show mucus production. MUC5A has been produced by cervical cells in a 2D culture setup even without flow.

As described earlier, the cervical epithelial cells in the cervical lumen experience mucus flow *in vivo* through periodic or continuous movement of mucosal discharges that can be regulated by different endocrine and microenvironmental stimuli. Our aim was to recapitulate the *in vivo*-like microenvironmental cues of the cervical epithelium, including the mucosal flow condition, leveraging the engineering design of the microfluidic chip device. While this might have been ignored in the past, application of flow was found to have significant influence on the epithelial cell phenotype and mucus composition as shown in **Fig. 2a** demonstrating the importance of these physical cues for cervix function. This is a major new finding in this study because this parameter could not be varied in past studies using static culture systems. Additionally, we showed that application of fluid flow resulted in significantly higher levels of the major cervical mucins (MUC5B and MUC4) than the static cultures (**Supp Fig. 1d**). MUC5A produced in 2D cultures is a minor product of cervical epithelium as MUC5B is the major mucin produced *in vivo*. So Transwells are not optimal models to study cervical mucus, nor do they have flow which is required for longer term co-cultures with microbiome that now can enable study of how mucus contributes to host-microbiome interactions. Finally, we did not only do biophysical analysis, we also carried out state-of-the-art glycomics analysis and confirmed that the chemical properties of the mucins produced more closely resembled mucus *in vivo* than that produced in Transwell cultures, which is another novel feature of our study. In sum, while certain mucins (e.g., MUC5A) have been previously shown in cultured cells, our data show that the mucus produced by the Cervix Chips more closely mimics that observed *in vivo* than mucus produced in past cultures based on both biophysical and biochemical analysis. This is a major finding of this study.

12. One of the major constituents of the cervical stroma are immune cells. Manuscript does not mention the contributions of immune cells. It is unclear how innate immune response is exclusively becomes the property of epithelial and stromal cells. This is a limitation of this model. The contribution of immune cells to biology is well reported in the cervical functions PMID: 24410939 PMID: 30629144 PMID: 28395330 PMID: 25811906 PMID: 19234164 PMID: 26637953. Stroma with macrophages has a well-defined role on microbial response and their proliferations and transitions are major contributors of cervical mucin biology, matrix remodeling and dysfunctions during dysbiosis.

The goal of our study was not to engineer a whole cervix. Rather, Organ Chip technology is a form of synthetic biology that allows researchers to control each potential control parameter (e.g., cellular composition, physical cues, chemical cues) independently, which enables one to get new insight into how organ-level biology and physiology are controlled. We agree that immune cells are an important player; however, our focus in this initial description of the Cervix Chip was on its use for studying mucus physiology *in vitro*. While immune cells can

likely influence cervical mucus in certain contexts *in vivo*, we decided to start with the key cellular components (epithelial and stromal cells) that have been shown in the past to be critical in early development, reproductive function and pathophysiology of this organ. Importantly, our results clearly show that immune cells are not required to replicate the physical and chemical features of cervical mucus seen *in vivo* as our Cervix Chip model can replicate many features of mucus biology as well as other features of cervical phenotype with greater fidelity than past *in vitro* models, even though immune cells are not present. However, immune cells can be integrated into these chips in the future, for example, if we set out to explore higher level responses to infection. In any case, we now describe this caveat and the ability to overcome this limitation in the future in the Discussion.

13. Please indicate the endogenous production and contribution of estradiol-17beta in your model or other endocrine factors. What is the need for exogenous supply?

The major source of fluctuating hormonal stimulation in the reproductive tract associated with the follicular and luteal phases is endocrine secretions from the ovary. Because we were interested in modeling the cervical tissue response to hormonal stimulations during the two phases of the cycle and we did not have the ovarian tissue in our model, we added estradiol and progesterone exogenously via inclusion in the medium that we perfused through the microfluidic channel of the chip. We now explain this more clearly in the manuscript.

14. Page 7; lines 144-145 – Why there is a decline in cell proliferation? How cellular stress of flow and or long term culture impact the cells?

The decrease in the cell proliferation is due to induction of differentiation in the cervical cells on-chip as described earlier in response to question #9. During the time that cell proliferation decreased, there was a concomitant increase in expression of the cell differentiation marker (i.e. MUC5B) indicating formation of more functional phenotype rather than cell death. In addition, quantitative measures of both proliferating and mucus producing cells reaches a plateau after 6 days that remains constant for the remainder of the culture period. Thus, neither long term culture nor application of flow is inducing progressive cell death in the epithelium on-chip.

15. Page 8; line 168 – is it possible that cytokines are washed out due to continuous flow? This is still not a physiologically valid model.

The reported cytokine levels were measured in the total accumulated chip effluents after 20 hours of static culture on-chip followed by 4 hours of flow to avoid any variability that may be induced by accumulation or washing off of the cytokines from the epithelial lumen due to changes in flow. Additionally, the exact volume of the chip effluents and epithelial cell number were measured for every chip to normalize the detected level of cytokines to ensure a fair comparison.

16. Page 10; If the ecto-cervical region is heavy in N-glycans does that mean it is not clinically and or physiologically relevant? Is this a limitation of the model? How does that translationally impact cervical function?

Currently, glycomic analysis of the clinical mucus can only reveal glycosylation type and structures that are present in vaginal secretions that contains a mixture of ecto- and endo-cervical mucus secretions as well as a lower level of mucus produced by the vaginal epithelium. This is because it is practically impossible to separately collect and analyze mucosal secretions from each of these regions clinically. As such, there is no clinical benchmark to compare the physiological relevance of our identified glycosylation patterns in the endo- and ecto-like Cervix Chips mucus. In fact, our model, has enabled, for the first time, glycomic analysis of mucosal secretions from Cervix Chips that have transcriptomic signatures similar to that of endo- and ecto-cervix, and revealed the different abundance of O- and N-glycosylated mucins in these two different epithelial phenotypes. Clinical cervicovaginal fluid is heavily O- and N-glycosylated which has been shown to be highly correlated with the state of host mucosal immunity, homeostasis, and microbial communities in pregnant and non-pregnant women. Cervix Chip models that enable teasing out of the unique contribution of each cervical sub lineages are highly relevant for understanding the physiology and pathophysiology of the lower reproductive tract in diseased or non-functional organs such as cervical cancer and fibrosis that can affect different parts of the cervix. In fact, some of the greatest interest we have obtained from OBGYN physicians and funding agencies (e.g., Gates Foundation) is precisely because we can study mucus production by cervical epithelium in the absence of confounding factors (e.g., variable microbiome, pathogens, hormonal status, sexual history, etc.) that complicates analysis of all clinical samples. We now more clearly explain this in the text.

17. The manuscript is largely and primarily focused on mucus production but relates poorly to cellular changes. Please provide emphasis to cellular changes as the multiplicity of cells makes cervix a unique organ and not just mucus.

As described above, we carried out multiple studies characterizing the cellular phenotype of the cervical epithelium cultured under two different flow conditions. This included immunohistochemical and transcriptomic analyses as well as characterization of mucus produced by these cells. However, a major advantage of our *in vitro* model is that it can allow us to study host-microbiome interactions, and in particular, how the cervical epithelium responds to healthy versus dysbiotic microbiome at the cell and molecular levels. Thus, to provide more emphasis on cellular changes, we carried out additional studies in which we performed proteomic analysis of the cervical epithelium on-chip 72 hours post co-culture with and without commensal (*L. crispatus*) versus dysbiotic (*G. vaginalis*) consortia, which we now include in **revised Fig. 4g**. These results showed that compared to control chips with no bacteria, colonization with *G. vaginalis* upregulates expression of several proteins in the cervical epithelial cells that are involved in the cell pathogenesis phenotype and processes including HLA-C (Class 1 HLA antigens) and TRIM25 (an E3 ubiquitin ligase enzyme) proteins regulating adaptive and innate immune responses, free ribosomal proteins (RPL15 and RPL30) participating in cellular development and cell cycling associated with cell apoptosis and tumor pathogenesis^{1,2}, ACADVL (very long-chain specific acyl-CoA dehydrogenase) protein that compromises tissue integrity and function through catalyzed mitochondrial beta-oxidation³, cells membrane vesicular trafficking protein (EXOC8) exploited and interfered by microbial pathogens for cell entry and infection^{4,5}, and enzyme transglutaminase 1 (TGM1) found in the epithelium outer layer suggesting the host defense mechanism to *G. vaginalis* by increased cell shedding⁶ (**Fig. 4g**)

Interestingly, colonization with *L. crispatus* significantly increased CSRP1 (cysteine and glycine-rich protein 1), 9PTGFRN (prostaglandin F2 receptor negative regulator), and CSTB proteins in the cervical epithelia cells compared to no bacteria condition, which are all highly expressed in the female reproductive tissues⁷ and are known to be involved in important cellular regulatory processes including cell development, differentiation, and protection against proteolysis enzymes^{8,9}, in addition to down regulating secretion of a free ribosomal protein (PRS28) (**Fig. 4g**). These data reveal the important role of commensal *L. crispatus* bacteria in regulating the cellular functions and responses of the host cervical epithelium as is observed in healthy cervix *in vivo*.

We also compared the differentially expressed proteins in *G. vaginalis* compared to *L. crispatus* colonized Cervix Chips and found that the CSRP1 and PTGFRN proteins are significantly downregulated showing the direct counter effect of dysbiotic compared to healthy commensal bacteria on epithelial cells. Furthermore, TRM25, RPL30, S100A7 (psoriasin), and ANKRD22 (ankyrin repeat domain-containing protein 22) proteins that aberrantly expressed in multiple types of cancer (i.e. cervical and ovarian cancer)¹⁰⁻¹² induces or induced by different proinflammatory cytokines and chemotaxis¹³ were found to be significantly upregulated in cervical cells of dysbiotic compared to *L. crispatus* Cervix Chips (**Fig. 4g**). These results are now included in the revised manuscript.

18. Page 14; line 322– such a flow of microbiome to impact endocervix does not exist in vivo. Endo is not in direct contact to the VEC or the microbiome. Ecto is and hence, the role of ecto to microbes and their repercussion on endo is more critical. See my comments above.

As described in our response to question #6, a recent study by Chen et al. 2017 published in this journal (*Nature Comm.*; DOI: 10.1038/s41467-017-00901-0) that systematically studied the microbiota composition of the female reproductive tract in 110 reproductive age women, clearly demonstrated the presence of commensal bacterial communities in cervical canal, uterus, fallopian tubes and vagina. Thus, both the endo- and ecto-cervix are in direct contact with microbiome.

19. Page 14; line 323 – "this would have the highest relevance for the health of the upper reproductive tract". This is not true. See my comments above. Please rewrite this sentence.

As we responded to questions #6 and 18, this Reviewer's belief that the endocervix is sterile is incorrect. More importantly, transmission of pathogenic bacteria from the vaginal canal to the upper reproductive organs is a major cause of prenatal labor and infant mortality. It occurs due to the pathogen's invasion and destruction of the mucosal barrier in the cervical canal, which is present in the endocervix. Because this is a critical part of the host defense against progressive infection that can negatively impact the health of the upper reproductive organs, we chose to model and study host-microbiome interactions in the endocervix chip with the thick mucosal layer. Our statement makes this point and emphasizes the importance of modeling this critical stage in health and disease state of the cervix. We have revised the statement in the manuscript to clearly reflect our rationale for modeling dysbiosis in the endocervix Chip.

20. Page 14; line 333 - It seems that 3 days is the limit for these cultures. The rest of the experiments are all 12 days. Please clarify as I am bit confused with the number of days in culture environment and for various experiments.

The Cervix Chip is developed and matured during a 12-day culture time including 5 days of expansion and 7 days of differentiation. After the Cervix Chip is matured and functionally stable at day 12 of culture, the chip was inoculated with the bacterial communities and co-cultured for additional 3 days. We have been able to co-culture the Cervix Chip for up to 5 days with the reported bacterial communities, but since the dysbiotic phenotype is detectable and characterizable at day 3 of co-culture, we performed and reported the data at day 3. We explain this more clearly in the revised Methods.

21. Fig 4; please explain if the vaginal microbiome interaction with vaginal epithelial cell has been factored in the discussion. These microbes, if any, interact with ecto cells before impacting endo cells. The metabolites of the microbiome – vaginal cell interactions produce most of the repercussion effect in subsequent upper genital tract layers.

We did not study the impact of ascending metabolites from vaginal infections; however, this could be studied in the future by fluidically linking the Cervix Chip with a Vagina Chip we described in a recent publication. Although this is beyond the scope of the present study that largely focuses on description of this new Cervix Chip model, we now highlight the importance of considering host-microbiome metabolites from vaginal cells in future studies in the Discussion.

22. Please include data related to collagen production by stromal and epithelial cells. ECM coating was mentioned (page 5) but the changes to this component was not mentioned in response to experimental conditions.

We have performed second harmonic imaging of the stromal layer on the Cervix Chip and observed accumulation of the collagen in the stromal layer by day 12 of culture. This image of the stromal layer is now included in the **Suppl. Fig. 1b** in the revised manuscript. Also, based on transcriptomic data, we identified the collagen isoforms that have been previously reported to be expressed at high levels in cervix *in vivo* using the GTEx database. We compared the collagen genes that are among the top 50% of expressed genes in the human cervix tissue with transcriptomic data from the Cervix Chips and found that the expressed collagen genes in the Cervix Chip are similar to those detected in endo- and ecto-cervical tissues *in vivo* as reported in GTEx data base. Additionally, the expression levels of these collagens were higher and more similar to *in vivo* levels in the Cervix Chips than the levels observed in static Transwell cultures (**Supplementary Fig. S2e**)

23. Authors should not try stratified analysis based on race with limited samples available. Without replication and validation, these data are merely misleading. The discussion on this topic should be avoided.

This is a helpful suggestion. The manuscript has been revised to remove any discussion based on the donor's race.

24. The comparison of cervical cell based data to airway epithelial cell is not a valid comparison. Please remove those references throughout the manuscript.

These references were removed from the revised manuscript.

25. Key references of people who contributed to the field of cervical biology (Mahendroo et al, Ward et al, Stock et al, Yellon et al) are missing. Their work should be recognized.

References to the relevant works from these groups are now included in the manuscript.

REVIEWER #2:

1. The rationale for mixing endo- and ecto-cervical epithelial cells is not provided. The endocervix and ectocervix are very different with different physiology. It was unclear why both were mixed together. Furthermore, the importance of the presence of stromal cells was not studied. Why were these cells included and what changes were observed in each of the experiments that were done? Given that continuous versus periodic flow rates promoted enrichment of different epithelial cells, all differences in functional measurements could simply be due to the different population of cells and not necessarily the impact of flow on the cells. Control experiments using just one cell epithelial cell type would promote a greater understanding of the impact of flow and shear stress.

We apologize for the confusion regarding the cells used in this study. A key goal of this study was to demonstrate the advantages it provides by benchmarking it against existing common *in vitro* organotypic cervical culture models. While most past cervical cultures used established cervical epithelial cell lines, more advanced versions used a commercial source of primary human cervical epithelial cells cultured in Transwell inserts (Zhang et al., *Oncol Lett.*, 2020; Caven et al., *Front Cell Infect Microbiol*, 2023). Thus, we chose to use the same commercial source of healthy primary human cervical epithelial cells that were used in these previous studies (which are only sold by two suppliers) for our studies. The vendors did not provide any technical information about the specific zone of the cervix from which the cells were isolated. In the course of characterizing these cells, we discovered that they contain a mixed population of both endo- and ecto-cervical cell types (presented in **Suppl. Fig. S1a**). We now explain this more clearly in the revised manuscript. However, we also discovered that we could push these cells preferentially down ecto- or endo-cervical differentiation paths using different flow conditions. This is, in itself, a novel finding of this work that could not be uncovered using conventional static systems. It is also important to note that purifying endo- and ecto- cervical cells from this mixed cell population for through sorting is not easily done. As these are primary cells, the cell yield after sorting is low and they have limited expansion capacity in 2D culture.

Epithelial-stromal (mesenchymal) interactions play a key role in development of the cervical epithelium and others have shown that the presence of stromal cells improves cervical

epithelial cell differentiation in static culture models. We confirmed this in preliminary studies in which we cultured cervical epithelial cells in the presence or absence of stromal cells in static Transwell culture. We found that stromal and cervical cells co-culture is essential for the formation of a uniform and electrically resistance cervical epithelium *in vitro*. In addition, we found that the stromal cells increased mucus production in the cervical epithelium. These data are now included in the **Supp Fig. 1c** of the revised manuscript. Given the finding that the presence of the stroma is critical for optimal differentiation and function of the cervical epithelium, studying the effects of fluid flow on isolated endo- or ecto-cervical epithelial cells is clearly beyond the scope of the present study.

2. Why were two different flow rates studied? Is this physiologically relevant? There is no continuous flow of liquid that the endo- and ecto-cervix is exposed to *in vivo*. Rather, mechanical stimulation to mimic uterine contractions would have been more physiologic.

Although human cervix is not perceived to experience a significant form of continuous flow similar to that in the intestine or arteries, mucus is continually produced, hydrated by interstitial fluids, and cleared from the surface of the epithelium as it moves (flows) to the vagina by mucociliary transport. This can be further augmented by gravity and by changes in endocrine hormones and other microenvironmental factors that influence mucus production in the reproductive tract. This flow, even if minimal, can influence epithelial cell function through associated application of fluid shear forces just as it does in many other organs. In a culture system such as this, the continuous removal of mucus, preventing it from accumulating abnormally, may be equally important to mimic the *in vivo* condition. As such, we believed that it is physiologically relevant that the cervical epithelium surface experience a gentle continuous or a less frequent periodic flow associated with propelling mucosal discharge. Because this was the first effort for developing the Cervix Chip and little is known about physiological mucus flow conditions *in vivo*, we did not know which flow condition would be optimal for promoting the growth and maturation of cervical epithelial cells. So, we tested both continuous and periodic flow conditions that mimic frequent and infrequent mucosal discharge flow, respectively, *in vivo*. This effort led to the interesting and novel observation that different flow conditions moved the mixed cervical epithelial population along different differentiation paths (ecto- vs. endo-cervical), which we report here. We now clarify our rationale more clearly in the manuscript.

Our work focuses on modeling the cervical epithelium rather than uterine epithelium. The cervix does not experience mechanical contraction forces except during the parturition, which was not the focus of this study.

3. *note that “ovulatory” vs “non-ovulatory” should not be used in this study to describe E2 vs E2+P4 treatment. Please use “follicular” for E2 treatment and “luteal” for E2+P4 treatment. Non-ovulatory is misleading and the field does not describe the hormonal stages of the cycle as such.

Thank you for your correction. These terms were corrected throughout the manuscript as suggested.

4. Throughout the manuscript the term “differentiation” is used. This implies that there are progenitor cells that differentiate to endo or ectocervix epithelium, and it is assumed

that this is not the case of the cell lines that you are using. You are treating the cells and the cells are responding to the stimulus. Cells are no differentiating. Please correct.

We are not using established cell lines; we are using primary cells. The term cell differentiation has different meanings for different biologists. We define it as many others do: "Cell differentiation is the process of cells becoming specialized in their structures and function and performing a certain job in the body." This is precisely what we show in this study in terms of changes in gene expression, mucus production, etc. associated with endo- and ecto-cervical epithelial cells. We now describe this definition when we first use the term in the manuscript.

5. The data showing race differences, Black, White and Hispanic is meaningless as N=1 for each race. Without sufficient power, it is not known whether race is the determining factor for different gene expression or whether it is reflecting the heterogeneity that naturally occurs between patients even within the same race. Furthermore, self reported race is not entirely accurate and at least for the Black population, ancestry analysis would confirm race. In addition, the race of the stromal cells was not determined.

Thank you for pointing this out. We have revised the manuscript to remove the donor's identification by the race.

6. As PDMS is known to absorb hydrophobic molecules, it would be important to measure E2 and P4 concentrations in the media after flowing through the device.

While PDMS is known for its capacity to absorb small, hydrophobic molecules such as estradiol, it can reach its absorption capacity after sufficient amount of the hydrophobic molecule has perfused through the channel. This creates a saturation equilibrium between the PDMS and the perfusing medium so a stable concentration of the molecule can be maintained in the chip (Toepke and Beebe, Lab Chip, 2006). We also performed functional assessments of the Cervix Chips after at least 7 days of continuous perfusion of hormone containing medium in the chip that assures saturation of the PDMS absorbance capacity and stable hormone level concentration on-chip. Finally, we performed studies in our chips using hydrophobic estradiol (E2) at ~200 pg/ml concentration in fresh medium flowing through the apical and basal channels of the chips for 48 hours and observed negligible loss in the E2 concentration in the chip effluents using mass spectrometric analysis of the chip medium in the inlet compared to the outlet chambers. We now clarify this in the Methods.

7. Line 24 – the authors state that they recreate the cervical epithelial-stromal but have no data to show the role of the stroma. How does the interface work mechanistically? After Fig 1F there is little to no reference to the contributions of the stromal cells to the correct functionality of the epithelial cells. Is there anything known about their use or function? It would be important to see the mucus production in periodic and continuous flow without the stromal cells present. Otherwise, this model could have been simplified.

As described earlier in response to your question #1, we performed studies to investigate the importance of stromal cells in the cervical epithelial culture using static Transwell system and found that stromal and cervical cells co-culture is essential for the formation of a uniform and electrically resistance cervical epithelium *in vitro*. We also observed that co-culture

of stroma with epithelial cells significantly increased secretion of mucus by the cervical epithelial cells. These data are now included in the **Supp Fig. S1c** in the revised manuscript.

8. Line 114 – Please explain which flowrates were used, ie, why only 40ul/h is optimal and why intermittent used 25% less flowrate for 4 hours.

Because this was the first effort at developing a Cervix Chip and little is known about physiological mucus flow conditions *in vivo*, we did not know which flow condition would be optimal for promoting the growth and maturation of cervical epithelial cells. So, we tested both continuous and periodic flow conditions that mimic frequent and infrequent mucosal discharge flow, respectively, *in vivo*. We chose 40 $\mu\text{l/hr}$ for continuous flow because that was previously found to support effective epithelial differentiation in a human Vagina Chip (Mahajan et al., *Microbiome*, 2022). We chose the lower flow rate (30 $\mu\text{l/hr}$) for 4 hrs followed by 20 hours of static culture to mimic more gentle mucus flow conditions, again because we did not know what might be optimal. However, this led us to discover that these different flow conditions push the epithelium down distinct ecto- vs. endo-cervical epithelial differentiation pathways. We now explain this rationale in the Methods.

9. Line 118 – Other systems did not continuously perfuse the stromal layer because these cells usually do not experience shear stresses – how is this 40ul/hr in a 1mm by 0.2mm channel physiological? What would happen with less flow in stromal cells?

It is important to note that compared to conventional systems, the dual channel microfluidic device offers the advantage of enabling long term culture and maintenance of the tissues functionality *in vitro* using controllable medium flow that replenishes nutrients and removes waste from the tissue microenvironment similar to that *in vivo*. This enhances differentiation and prevents tissue cell death that can sometimes lead to short culture times in conventional static cultures. We previously showed that use of the same flow rate (40 $\mu\text{l/hr}$) through a microfluidic channel lined with stroma cells within a human Vagina Chip resulted in high levels of epithelial differentiation and mimicry of human vagina physiology and pathophysiology (Mahajan et al., *Microbiome*, 2022). This flow rate produces a low level of shear stress (0.0003 dyne/cm² in the apical channel and 0.012 dyne/cm² in the basal channel); however, the continuous flow enabled formation of a thick stromal layer that experienced diffusion of nutrients to the overlying cervical epithelium, which is similar to what the stromal and epithelial cells experience *in vivo*. We also tested using a lower flow rate in the stromal channel and observed increased growth of the stromal layer and blockage of the basal channel that compromised medium flow.

10. Fig 1A – Add the term apical and basal to image.

The image was revised.

11. Fig 1D – Indicate the confines of the channel.

The image was revised.

12. Line 132-138 and Fig 1E-D: The authors have done a good job demonstrating that the

cervical epithelial cell layer presents cervical epithelium-specific markers, however no comment on the morphology of the construct is made. The concern here is that they used a commercial mix of epithelial cells from the endo- and ectocervix, which have very different morphologies in vivo (stratified squamous and columnar mucinous epithelium respectively). From the images it looks like the epithelial construct is only a couple of cell layers thick with no tissue-specific morphology. Please elaborate in the manuscript.

We performed Pseudo H&E staining of the Cervix Chips and observed that the epithelial cells under continuous flow condition forms a stratified epithelium with multiple (> 6) layers of flattened cells that is reminiscent of ectocervix. Under periodic flow, the epithelial cells formed fewer layers (2-4) and exhibited a range of forms from cuboidal to more flattened. Thus, while the morphology of these cells cultured under periodic flow did not fully replicate the columnar form of the endocervix, these flow conditions did seem to prevent ectocervix-like stratification as observed under continuous flow. These images are now shown in **Supp Fig. S1b**.

13. Line 148 & Fig s1d – is this different expression observed in both flow conditions? Are the intermittent and continuous results averaged?

The data presented are from the periodic flow chips. This is now clarified in the revised Results and figure legend.

14. Line 158 & Fig s1e – Barrier function was measured using TEER and chip vs Transwell resistance was measured. Can you elaborate on what kind of Transwell was used, and if it also had an adherent layer of epithelial cells and stromal cells divided by a permeable membrane? This information is also useful to add to the methods in line 479. Lastly, assuming that the cell numbers and surface areas are very different between transwells and chips, is there a type of normalization necessary to be able to compare these cultures?

The information regarding the Transwell culture set up is now included in the Results and Methods section. The cervical epithelial cell and stromal cell seeding densities in the Transwell culture were adjusted to be the same as in the Organ Chips. We also normalized the TEER levels in both Chip and Transwell cultures to the surface area to enable fair comparison of the TEER values. This is also now clarified in Methods in the revised manuscript.

15. Line 168 – is the decrease in IL-8 and TNF- α is due to less mechanical stimulation or due to the accumulation of waste products during the 20 hour static period.

Cytokine analysis was performed on the chip effluents collected after 24 hours for both the periodic and continuous flow chips to ensure that the accumulation of waste products does not influence the cellular readouts of the secreted cytokines. We also measured the exact volume of the chip effluents and cell number/chip to normalize the concentration of detected cytokines so that their levels can be fairly compared across the two conditions. Previous studies have reported that cyclic mechanical stimulation (for example, those occurs during labor), significantly elevates the IL-8 secretions in uterine epithelial cells in addition to the known induction of IL-8 production through TNF- α stimulation (Takemura et al., *Molecular Human Reproduction*, 2004; Osawa et al., *Infection and Immunity*, 2002; Dunican et al., *Shock*, 2000).

Interestingly, the periodic flow chip (with endo-cervical transcriptome phenotype) showed a similar response to the cyclic flow stimulation on-chip. We have now clarified this point in the revised Results.

16. Line 170 – Given the different cell populations, it would have been more informational to have done single-cell RNA sequencing and demonstrate/cluster the survival of specific cell populations in periodic or continuous flow. Is there any way to isolate relative quantities of endo and ectocervical epithelial cells from this data?

While this study would be interesting, it would be difficult given the low numbers of the different cell types in these chips. More importantly, this is beyond the scope of this study which focuses on the initial description of an improved *in vitro* model of the human cervix that can be used for studies of cervical mucus physiology and host-microbiome interactions.

17. Line 221 – this is the first time that endo- and ecto-chip has been used, and should be defined here.

This was corrected in the manuscript.

18. Line 237 – Is the clinical mucus considered as a mixture of endo and ecto-cervix excretions?

Yes, the clinical mucus collected from the donor is a mixture of endo- and ecto-cervical secretions, as well as low levels of vaginal secretions. This information is now included in the revised manuscript.

19. Line 241 – Please add that permeability plays a critical role during reproduction and allowing the sperm through.

Thank you for the comment. This was added with relevant reference.

20. Line 302 – Does a decrease in these N-glycans have an effect on the mucus permeability, has that been described?

The specific effect of N-glycans in the mucus permeability has not been described previously. However, decreased abundance of negatively charged (hydrophilic) glycosylation groups lowers mucus hydration and results in formation of a dense mucosal layer that can reduce mucus permeability, as we observed in the Cervix Chip treated with the luteal phase hormonal condition.

21. Line 325 – Please describe why estrogenic conditions were used.

The follicular phase of the menstrual cycle with higher level of estradiol has shown to strongly associate with a “healthy” state and a more stable *L. crispatus* dominated microbiome in the female reproductive tract. It is important to model and study the host mucosal microenvironment in this competent hormonal phase *in vitro* to better understand how dysbiotic bacteria initially invade and destruct the host defense mechanism in a healthy state. This is now explained in the manuscript.

REVIEWER #3:

1. The title implies there were two kinds of chips develop: one with endo-cervical cells and one with ecto-cervical cells and this is misleading. Better to simplify it as cervix chips. Alternatively, periodic and continuous flow can be integrated to the title.

Thank you for your suggestion. We revised the title.

2. How were exo and endo cervical cells mixed? Same ratio? Why were they mixed instead of colonizing different parts of the chip? What was the reasoning for not studying them separately? Please justify the decision. Why only cervical stromal fibroblasts isolated from the donor and not the cells to match the donors?

The ecto and endo cervical cells were not intentionally mixed. A key goal of this study was to demonstrate the advantages it provides by benchmarking it against existing common *in vitro* organotypic cervical culture models. While most past cervical cultures used established cervical epithelial cell lines, more advanced versions used a commercial source of primary human cervical epithelial cells cultured in Transwell inserts (Zhang et al., *Oncol Lett.*, 2020; Caven et al., *Front Cell Infect Microbiol*, 2023). Thus, we chose to use the same commercial source of healthy primary human cervical epithelial cells that were used in these previous studies (which are only sold by two suppliers) for our studies. The vendors did not provide any technical information about the specific zone of the cervix from which the cells were isolated. In the course of characterizing these cells, we discovered that they contain a mixed population of both endo- and ecto-cervical cell types (presented in **Suppl. Fig. S1a**). We now explain this more clearly in the revised manuscript. Cervical stromal cells are not available commercially and had to be isolated from human tissue to create the Cervix Chip. Unfortunately, healthy cervical tissues from different donors are not easily available for primary epithelial cell isolation, and we needed to rely on limited primary healthy cervical stromal cells we isolated for establishing the Cervix Chip model in this study. We have revised the manuscript to clearly explain our choice of using commercial cells source with mixed population in this study.

3. Lines 97-98. The statement to describe endo and ecto cervix for non-experts can be introduced earlier in the introduction. Where those cells are located, their function, and relative importance.

Thank you for your comment, we have included brief description on endo and ecto cervix in the introduction.

4. Line 79 and Lines 373-374. The 3D models such as RWV can capitulate the barrier properties and mucus production as well, so this statement needs to be corrected.

Although the Rotating Wall Vessel (RWV) has been shown to enable cervical cell aggregate formation and some biological responses to bacterial insults, the epithelial-stromal interface barrier, mucus production functions, and ability to study host-microbiome interactions over time cannot be physiologically modelled in the RWV system. This is now discussed in the Introduction.

5. Line 158: please also report the electrical resistance of transwell cells to see the magnitude for the comparison

Thanks for pointing this out. The measured TEER value for the Transwell was added to the revised Results text section.

6. Line 170: hyphen between RNA and seq

This was corrected.

7. Line 180: briefly described the differences observed on endocervical features

The referred features are now briefly described in this revised section.

8. There is a limitation based on lack of biological replicates of donor samples (based on ethnicity) so any comparison to the in vivo is premature. The trends observed can be only defined as a case study. Please tone it down the implications.

Thank you for your comment. We revised the wording to remove discussion of ethnicity.

9. Figure 1h. the bar graphs do not align. How many chips were used for each condition? Is it three? Please include on the figure legend.

This is corrected in the revised manuscript. The number of chips used in each of the figures is now included in every figure legend.

10. Figure 2b: the top of the boxes for O-glycans and N-glycans do not align

The tables have been revised and now align.

11. Line 1002: What does biological chip replicate mean? Does it mean different donor? or are they experimental replicates?

It means experimental replicates. We revised the wording to clarify this throughout the manuscript.

12. Figure 3. The microscopy images lack the scale information.

The scale information of the scale bars in panel a, d, and g are described in the legends of Fig 3.

13. Line 714. The sentence ends abruptly, maybe also mention the author name at the end.

This was revised and the author's name was added to this sentence.

REVIEWERS' COMMENTS

Reviewer #2 (Remarks to the Author):

Even though there are a number of new discoveries made in this study, there is still confusion as to whether this is a reliable physiological in vitro cervix model and which cervix this is modeling. The authors state that their goal was not to create an artificial cervix, but then also state that their goal is to "develop more clinically relevant in vitro model of mucus production, hostmicrobiome interactions, hormone sensitivity, and innate immune responses in human cervical tissues than is currently available today". This requires a clear delineation of whether the cervix is endo- or ecto- in origin. The use of a mixture of endo- and ecto-cervical epithelial cells is problematic.

Reviewer #3 (Remarks to the Author):

The authors sufficiently addressed my concerns.

Reviewer #4 (Remarks to the Author):

This manuscript represents a reliable in vitro model of different mechanisms governing the human cervical tissues; the manuscript provides sufficient data to prove that the "cervix chip" can be used to study mucus production, host microbiome

interactions, hormone sensitivity, and innate immune responses using commercially available cells from the cervix, and a specific hormonal and fluid-mechanical stimulation to control cell differentiation.

Based on these points, the model is significant in the field, as it overcomes limitation of current static, 2D approaches.

The mucous production by the epithelial cells and the characterization of this in response to different culture protocols/stimuli to the cells (i.e. flow) complete the model. The model is also used to observe the response of healthy and dysbiotic microbiome to bacteria and the microfluidic characteristics of the model are key and perfect to mimic a dynamic environment, more physiologically relevant than monodirectional, static cultures.

The methods and the results are complete and benchmarked to traditional transwell which is the standard method for the proposed assays. The possibility to guide the differentiation of the epithelial

cells is an interesting concept that could be applied to other organs. Future experiments should be dedicated to use the model with patient derived samples, considering age, hormonal stimulation and genetic characteristics to classify the samples.

RESPONSE TO REVIEWERS

REVIEWER #2:

1. Even though there are a number of new discoveries made in this study, there is still confusion as to whether this is a reliable physiological in vitro cervix model and which cervix this is modeling. The authors state that their goal was not to create an artificial cervix, but then also state that their goal is to "develop more clinically relevant in vitro model of mucus production, host microbiome interactions, hormone sensitivity, and innate immune responses in human cervical tissues than is currently available today". This requires a clear delineation of whether the cervix is endo- or ecto- in origin. The use of a mixture of endo- and ecto-cervical epithelial cells is problematic.

We have provided extensive experimental data and compared our results with benchmarked 2D and static culture models to demonstrate that the Cervix Chip model more closely mimics human cervical physiology, including mucus production, host microbiome interactions, hormone sensitivity, and innate immune responses than conventional culture systems, and this was recognized and appreciated by the other Reviewers. Most of the physiological functions modelled on-chip are common between endo- and ecto-cervix and clear delineation of these by the cervix chip phenotype does not compromise the fundamental relevance of this preclinical chip model. Nevertheless, we now extensively discuss the limitation of using a commercially available cell source with mixed cell types in this study and suggest that to strengthen the model use of primary epithelial and stromal cells from each of the endo- and ecto- distinct regions of the human cervix should be further investigated in the future.